# Oncogenic structural aberration landscape in gastric cancer genomes

**Mihoko Saito-Adachi** [1], **Natsuko Hama**[1], **Yasushi Totoki** [1,2], **Hiromi Nakamura**[1], **Yasuhito Arai** [1], **Fumie Hosoda**[1], **Hirofumi Rokutan** [1,3], **Shinichi Yachida** [1,2], **Mamoru Kato** [4], **Akihiko Fukagawa**[1] & **Tatsuhiro Shibata** [1,5] ✉

Structural variants (SVs) are responsible for driver events in gastric cancer (GC); however, their patterns and processes remain poorly understood. Here, we examine 170 GC whole genomes to unravel the oncogenic structural aberration landscape in GC genomes and identify six rearrangement signatures (RSs). Non-random combinations of RSs elucidate distinctive GC subtypes comprising one or a few dominant RS that are associated with specific driver events (*BRCA1/2* defects, mismatch repair deficiency, and *TP53* mutation) and epidemiological backgrounds. Twenty-seven SV hotspots are identified as GC driver candidates. SV hotspots frequently constitute complexly clustered SVs involved in driver gene amplification, such as *ERBB2*, *CCNE1*, and *FGFR2*. Further deconstruction of the locally clustered SVs uncovers amplicon-generating profiles characterized by super-large SVs and intensive segmental amplifications, contributing to the extensive amplification of GC oncogenes. Comprehensive analyses using adjusted SV allele frequencies indicate the significant involvement of extra-chromosomal DNA in processes linked to specific RSs.

Gastric cancer (GC) is the fifth most common cancer worldwide. Various genetic and epigenetic alterations involving GC drivers, such as *ERBB2* and *VEGFA*, have been recognized as key driver events and represent important therapeutic targets[1,2]. Comprehensive genomic analyses have uncovered diverse driver events and mutational processes defined by mutational signatures characterized by geographical, epidemiological, and histological backgrounds[1,3,4]. In contrast, the understanding of the oncogenic processes via structural variants (SVs) is currently being developed and some key driver SVs have been reported, such as *CLDN18-ARHGAP* fusion gene[5] and *KLF5* enhancer amplification[6]. Recent studies have explored patterns of structural alterations (SV signature[7] or rearrangement signatures[8]) across multiple human cancer types. Nevertheless, the etiologies of these patterns remain elusive because these studies did not compare the clinical and epidemiological backgrounds.

Furthermore, other studies have reported multiple types of complex or clustered SVs in cancer genomes[9,10]. Amplification is a genetic hallmark of activated oncogenes, where locally clustered SVs are frequently involved[9,11,12]. One molecular mechanism that leads to drastic segmental amplification involves extrachromosomal DNA (ecDNA), which originates from the inappropriate severing of chromatin bridges and subsequent chromosome fragmentation[13,14]. Subsequently, one or more of these fragments engage in DNA repair and generate circular structures[15]. Within the circular DNA structure, accumulation of subclonal SVs has been observed during cell division, resulting from circular recombination and error-prone repair[16]. These ecDNAs are unequally segregated into daughter cells, increasing the oncogene copy numbers efficiently and heterogeneously[17]. Esophageal adenocarcinoma studies also reported an association between clustered rearrangement signatures and complex SVs including ecDNA[18].

[1]Division of Cancer Genomics, National Cancer Center Research Institute, Tokyo, Japan. [2]Department of Cancer Genome Informatics, Graduate School of Medicine, Osaka University, Osaka, Japan. [3]Department of Pathology, Graduate School of Medicine, The University of Tokyo, Tokyo, Japan. [4]Division of Bioinformatics, National Cancer Center Research Institute, Tokyo, Japan. [5]Laboratory of Molecular Medicine, The Institute of Medical Science, The University of Tokyo, Tokyo, Japan. ✉e-mail: tashibat@ncc.go.jp

In this study, we aim to explore the structural rearrangement landscape of oncogenes in GC genomes. The non-random combinations of SVs identified by examining 170 GC genomes elucidate distinctive GC subtypes associated with epidemiological and driver backgrounds, histogenesis, and mutational signatures. We further deconstruct the locally clustered SVs (SV clusters) and uncover characteristic and heterogeneous amplification processes involving GC oncogenes.

## Results

### Whole genome mutational signature analysis of 170 GC genomes

We performed a whole-genome sequencing (WGS) analysis of 81 Japanese GC cases[4]. By combining the data with previously deposited GC WGS data[19], 170 GC WGS data were analyzed using the same pipeline (Supplementary Data 1). This identified 5,376,590 single-nucleotide variants (SNVs) and 26 single-base substitution signatures (SBS) (Supplementary Fig. 1). Chromatin status significantly affected the frequency of the GC mutation signatures. In the active-chromatin area, SBS1 (clock-like), SBS3 (*BRCA1/2*), SBS5 (clock-like), SBS6 (defective mismatch repair (*MMR*)), SBS13 (*APOBEC*), and SBS16 (alcohol-associated) were predominant. Furthermore, SBS8 (unknown cause), SBS9 (polymerase eta somatic hypermutation activity), SBS17 (unknown cause), and SBS18 (damage by reactive oxygen species) were predominant in the inactive area (Supplementary Data 2, Supplementary Fig. 1a). Of these, seven groups (SBS3, 6, 16, 17-1, 17-2, 18, and 28) exhibited distinctive contributions to SBSs (Supplementary Fig. 1a). These groups included two SBS17-dominant groups (SBS17-high and -low), of which the SBS17-high cases exhibited poor prognoses (Supplementary Fig. 1b). Additional properties of SBS and other mutational signatures are presented in Supplementary Data 3 and Supplementary Fig. 1.

### Global SV landscape in GC genomes

We developed an in-house SV caller (callallSV) to detect SVs that are not biased toward SV types and minimize false negatives, whose internal parameters were determined by a validation experiment involving 123 SVs (see **Methods**) and verified in a recent study[20]. The pipeline had two algorithms, paired-end (PE) and soft-clip detections, which were executed independently. Subsequently, the detection results of both were merged (see **Methods** and Supplementary Fig. 2). Based on the ratio of the number of reads supporting SV to the reference genome, we also calculated the adjusted SV allele frequency (SVAF) (see **Methods**).

A total of 49,059 SVs were identified, including 22,179 deletions, 11,234 tandem duplications, 8534 inversions, and 7112 translocations in the 170 GC genomes (Supplementary Data 4 and Supplementary Fig. 3). Deletions had a trimodal (35 bp, 1.4 kb, and 0.14 Mb) size distribution, which was similar to that observed in a previous pan-cancer WGS study[7]. In contrast, tandem duplications showed a bimodal (45 bp and 0.25 Mb) distribution (Fig. 1a). Small-scale (~100 kb) SVs involving deletion and tandem duplication events showed a higher SVAF, indicating their early acquisition. Tandem duplications with high copy number changes (ΔCN) revealed a significantly higher SVAF with a size-dependent tendency, indicating their potential as cancer drivers (Fig. 1a right, *P*-value < 0.001 of permutation test). The average translocation SVAFs were low compared to those of other types, confirming that they were acquired by genome-wide hypomethylation during cancer progression[21]. Six hundred fifty-two SVs were predicted to generate the fusion genes (Supplementary Data 4). Compared with the RNAseq data available in 62 cases (Supplementary Data 1), 24.5% of them (54/220) were expressed. Intragenic SVs with high SVAFs contained driver suppressor genes, such as *TP53* and fusion target genes, such as *ARHGAP26* (Supplementary Figs. 4a and 5a, b). High SVAF tandem

duplications contained driver oncogenes such as *CCNE1* and *MYC* (Supplementary Figs. 4b and 5c).

### Recurrent SV hotspots represent GC driver candidates

Using SVAF, we sought to identify GC-driver SVs genome-wide. Hotspots of high-SVAF intragenic SVs were frequently located within the GC tumor suppressor gene and fusion gene loci (Supplementary Fig. 6 and Supplementary Data 5). Meanwhile, hotspots of high-SVAF tandem duplication and inversion existed in known oncogene loci (including *ERBB2* and *CCNE1*), as well as in other loci. We focused on these two SV types and identified 27 genomic segments as candidates for driver SV hotspots that satisfied the following four conditions:1) detected in > 5% of cases, 2) not common fragile sites (CFSs) or active transposons, 3) containing five or more high-SVAF (> 0.4) cases, and 4) cases with SVs showing significantly increased copy number ratios than cases without SVs (Fig. 1b). These criteria covered well-validated GC oncogenes (Supplementary Data 6); furthermore, we also discovered genomic regions, as described below.

These hotspots were divided into singular- and multiple-type hotspots. The singular-type hotspot has a narrow region with rearrangements, shared among 70% or more cases and one or few SVs usually occurring in each case. For example, a duplication of the chr16:11,891–11,916 kb genomic region containing *BCAR4* occurred in 13 cases (Fig. 1c). *BCAR4* encodes a long non-coding RNA and is reported to be associated with malignant potential[22]. *BCAR4* knockdown significantly reduced GC cell proliferation (Supplementary Fig. 7). Other singular-type hotspots include previously reported duplications of super-enhancer segments of *KLF12-KLF5* (13q22) and *ZFP36L2* (2p21), resulting in their overexpression[6,23].

Multiple-type hotspots usually involve several genes and more than one type of SV. For example, tandem duplications or inversions occurred within a 20q13 segment (Fig. 1d, left), and four genes, including *SNAI1* and *CEBPB*, exhibited increased copy number and gene expression (Fig. 1d, right). *CEBPB* is a transcription factor that regulates IL6/8 cytokines in inflammatory responses and angiogenesis[24] and has been shown to regulate GC cell proliferation using the DepMap RNAi database (DepMap[25,26] public 21Q2). We also confirmed that RNAi knockdown of the *CEBPB* gene significantly reduced cell viability in two GC cell lines, MKN45 and HGC27, both of which had *CEBPB* copy gains (Supplementary Fig. 7).

### Rearrangement signatures (RSs) and subtypes in GC

To explore the molecular and epidemiological background of SV accumulation processes, we extracted SV signatures representing SV propensity. Previous studies have reported two methods of classifying SVs: one proposed by the Pan-Cancer Analysis of Whole Genomes[7] and the other by Signal[27] (https://signal.mutationalsignatures.com/). In this study, we used 80 features of SV classification based on the latter schema with additional categories (**Methods**, detailed comparisons of the three methods are shown in Supplementary Data 7): small-sized SVs, transposition-type, and chromatin states that are significantly associated with the distribution of mutational signatures (Supplementary Fig. 1). We extracted six structural rearrangement signatures (RS), each of which presented a characteristic chromatic distribution (Fig. 2a and Supplementary Fig. 8). All RSs, except RS1, exhibited higher frequencies in the active chromatin regions, which was pronounced in RS3 and RS4 (Supplementary Fig. 9). We compared these six RSs with previously reported reference signatures of rearrangement (RSR) for stomach cancer[27] (Supplementary Fig. 10d, e). Significant correlations were observed between RS3 and RSR1 (τ = 0.60); RS4 and RSR7 (τ = 0.64); and RS6 and RSR6b (τ = 0.55). RS1 and RS5 showed relatively weak correlations with RSR9 (τ = 0.20) and RSR4 (τ = 0.41). No RSR corresponded with RS2, which was characterized by the co-occurrence of small (<1 kb) deletions and tandem duplications (Supplementary Fig. 11).

Unsupervised hierarchical clustering classified 170 cases into seven RS subtypes (subtype RS1–6 and subtype RS2/6); six comprised each RS as a dominant signature, and the last one comprised a mixture of RS2 and RS6 (Fig. 2b).

Deletion-type SVs were predominant in RS1 and RS4 (Fig. 2b). RS1 was mainly composed of small-sized deletions (<1 kb), which spread genome-wide and were independent of the common fragile sites (Supplementary Fig. 12). This feature was similar to the SV feature of *BRCA*-altered breast cancers[8], and consistently, GC cases in the subtype RS1 had a significantly higher contribution of SBS3 ($P = 3.9 \times 10^{-2}$,

Welch's t-test), small insertions and deletions signature 6 (ID6) ($P = 5.6 \times 10^{-3}$), and ID8 ($P = 4.2 \times 10^{-3}$), all of which were associated with defective homologous recombination-based repair (Supplementary Fig. 10). *Omikli*, diffuse hypermutation induced by *APOBEC3* activity[28], was positively correlated with RS1 (Supplementary Data 8) and was frequently observed in subtype RS1 (Table 1). In contrast, RS4, consisting of large (0.1 to 1.0 Mb) deletions, showed a negative correlation (−0.261) with *omikli* occurrences (Supplementary Data 8). Cases in subtype RS4 significantly lacked whole-genome duplication ($P = 1.0 \times 10^{-12}$) (Table 1).

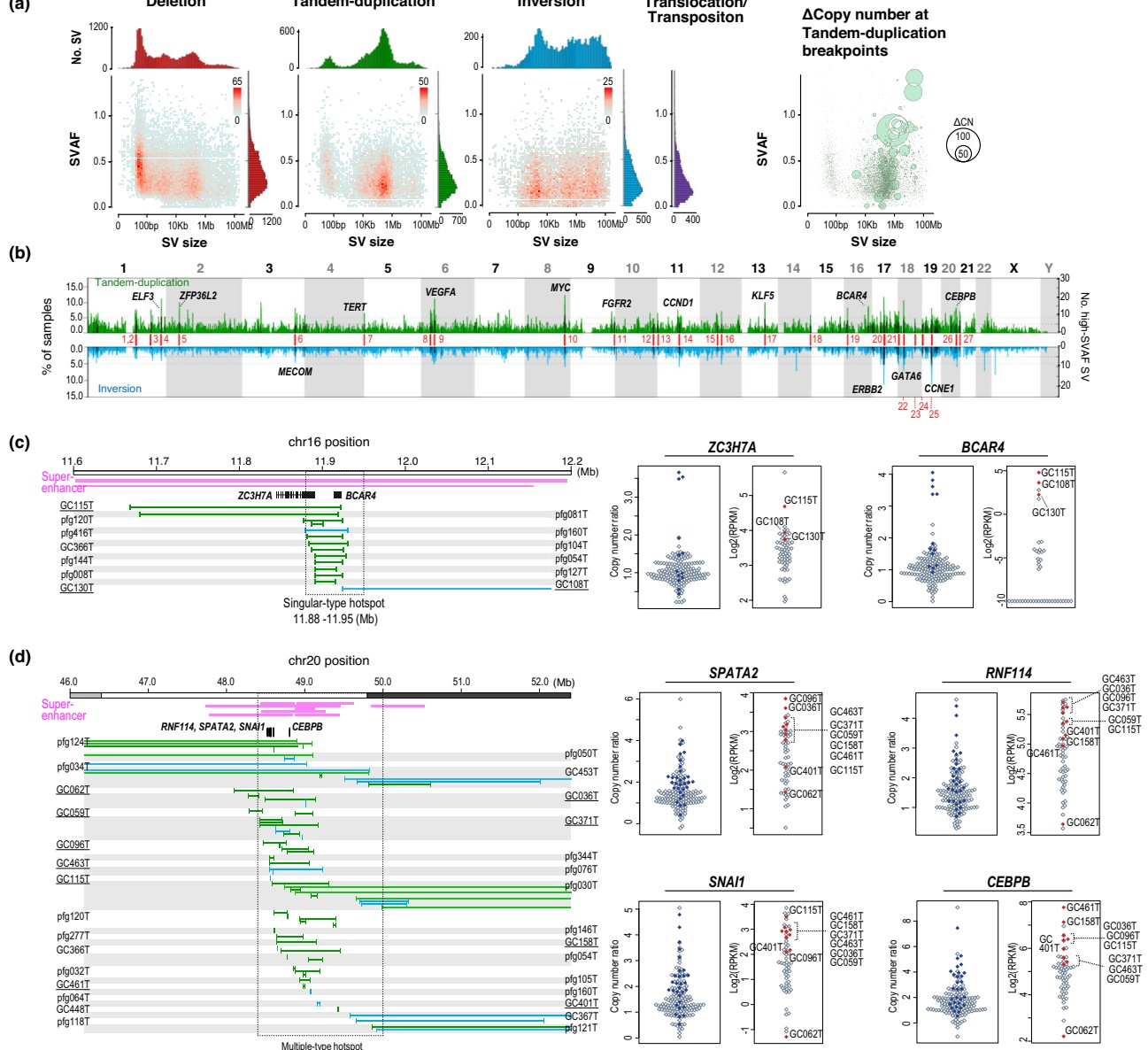

**Fig. 1 | Overviews of structural variants in 170 gastric cancer (GC) genomes.**
**a** The upper panels show the size distribution of structural variants (SVs). The middle panels show SV size and SV allele frequency (SVAF). The color of each dot represents the number of SVs. In the right panel, the copy number change at the breakpoints of tandem duplications is represented by the bubble size.
**b** Genome-wide frequencies of tandem duplications (green) and inversions (cyan). The histograms depict a sliding window of 500 kb (50 kb overlap), and the Y-axis shows the frequency of the cases. The black histograms indicate the number of SVs with SVAF > 0.4. See Supplementary Fig. 5 for all four SV types. **c** Example of a singular-type SV hotspot on chromosome 16. The upper panel indicates the distribution of super-enhancers (pink) defined by the ROSE

algorithm[50] from H3K27Ac peaks of gastric cancer cell lines ($n = 5$)[49], and tandem duplications (green) and inversions (cyan) at the chromosome 16 locus detected in these cases. The sample identifiers with underline indicate that they have transcriptional data (RNA-seq). The bottom panel shows copy numbers and expression (RPKM) of *ZC3H7A* and *BCAR4*. Each dot represents a case, and blue (copy number plot) and red (RPKM plot) indicate hotspot-positive SV cases. **d** Example of a multiple-type SV hotspot on chromosome 20. As shown in **c**, the upper panel indicates the distribution of super-enhancers and SVs in each case. The bottom panel presents the copy number and expression (RPKM) of four genes, including *CEBPB*, located within the hotspot. **a**–**d**, Source data are provided as a Source Data file.

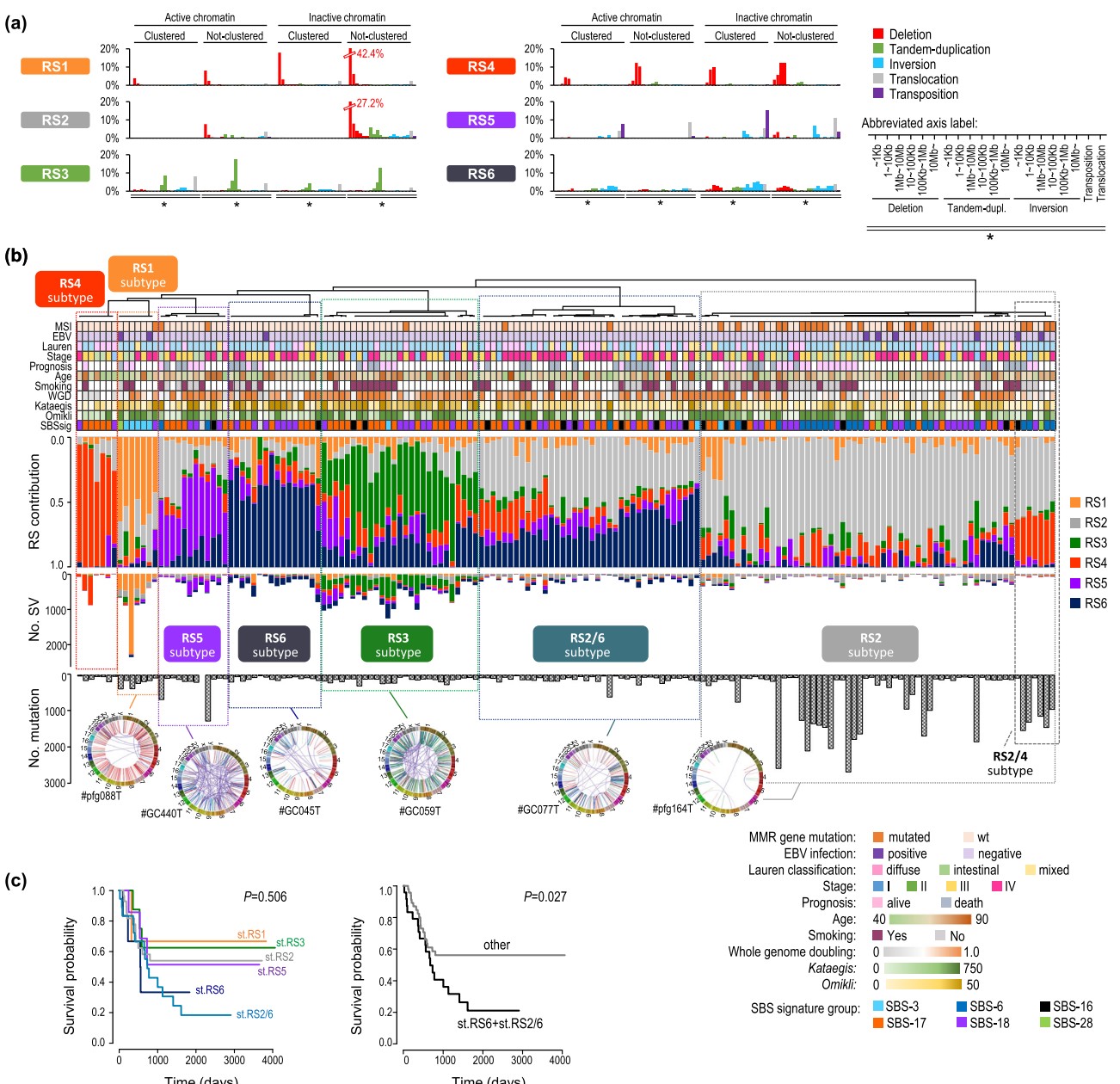

**Fig. 2 | GC Structural rearrangement signatures (RSs). a** Six RSs (RS1–RS6) were extracted using non-negative matrix factorization. The stability of the RS is shown in Supplementary Figure 10. The X-axis shows 80 categories by combining SV size (<1 kb, 1–10 kb, 10–100 kb, 100 kb–1 Mb, 1–10 Mb, > 10 Mb), SV type, distribution (clustered or non-clustered), and chromatin status (active or inactive). The Y-axis indicates the probability for each category. The color bars denote the SV types: deletion, red; tandem duplication, green; inversion, cyan; translocation, light grey; and transposition, purple. **b** Unsupervised hier-archical clustering of 170 cases based on RS contribution elucidated seven RS subtypes (abbreviated as st.RS1–st.RS6, and st.RS2/6 in Figures). Clinical information and genomic features are shown at the top of the Figure. If no data is available, a field is left blank. The RS contributions and numbers in each case are shown by the colors (orange, RS1; light grey, RS2; green, RS3; vermillion, RS4; purple, RS5; indigo, RS6). The total number of single nucleotide mutations is shown at the bottom of the Figure. Representative Circos plots for each RS subtype are shown. **c** Kaplan–Meier survival analysis of the RS subtypes. The left panel shows the survival curves of seven RS subtypes, and the right panel presents the survival curves of RS6-dominant subtypes (subtypes RS6 and RS2/6) compared with other subtypes. P-values are calculated using the log-rank test. **a–c**, Source data are provided as a Source Data file.

RS3 is characterized by a large (1 kb–1 Mb) tandem duplication in the active chromatin area. Tandem duplications are frequently accompanied by an increased copy number. Cases of subtype RS3 were associated with older age ($P = 5.2 \times 10^{-3}$), smoking status ($P = 4.1 \times 10^{-3}$), and the presence of *TP53* mutations ($P = 7.3 \times 10^{-6}$). This subtype also presented a higher number of SVs ($P = 7.2 \times 10^{-9}$), whole-genome duplication ($P = 4.4 \times 10^{-4}$), and *kataegis* events ($P = 3.0 \times 10^{-5}$) (Table 1).

RS2 and RS6 exhibited contrasting SV features: RS2 comprised sparsely distributed small SVs, wherein RS6 comprised densely distributed large SVs. Subtype RS2 included hyper-mutated cases ($P = 3.5 \times 10^{-8}$), and their SV frequency was low ($P = 5.0 \times 10^{-9}$). This subtype showed significantly higher con-tributions of SBS6 ($P = 1.0 \times 10^{-5}$), SBS12 ($P = 1.7 \times 10^{-5}$), ID7 ($P = 4.1 \times 10^{-3}$), and doublet base substitutions (DBS) signature 10 ($P = 6.0 \times 10^{-3}$) (Supplementary Fig. 10), all of which are

**Table 1 | Clinical and molecular backgrounds of the RS subtypes**

| subtype | st.RS1 | st.RS2 | | st.RS3 | | st.RS4 | | st.RS5 | st.RS6 | | st.RS2/6 | |
|---|---|---|---|---|---|---|---|---|---|---|---|---|
| Characteristic SV features | Small-deletion | Sparse-small SVs | | Tandem-duplication | | Large-deletion | | Translocation | Dense-large SVs | | Both RS2 and RS6 dominate | |
| No. member | 7 | 61 | | 27 | | 7 | | 12 | 16 | | 38 | |
| No.SV | - | $5.0\times10^{-9}$ | ▼ | $7.2\times10^{-9}$ | △ | - | | - | - | | $1.2\times10^{-4}$ | ▼ |
| No. mutation*1 | - | - | | - | | - | | - | $4.6\times10^{-2}$ | ▼ | - | |
| WGD | - | $1.2\times10^{-2}$ | ▼ | $4.4\times10^{-4}$ | △ | $1.0\times10^{-12}$ | ▼ | - | - | | - | |
| SCNA | - | - | | $4.1\times10^{-2}$ | △ | - | | - | - | | - | |
| Diffuse type | - | - | | - | | $1.7\times10^{-2}$ | △ | - | - | | $7.7\times10^{-4}$ | △ |
| Intestinal type | - | - | | $9.1\times10^{-3}$ | △ | - | | - | - | | $2.3\times10^{-4}$ | ▼ |
| Smoking | - | - | | $4.1\times10^{-3}$ | △ | - | | - | - | | - | |
| Age | - | - | | $5.2\times10^{-3}$ | △ | - | | - | - | | - | |
| Stage*2: I/II | - | - | | - | | - | | - | $1.3\times10^{-2}$ | ▼ | $3.0\times10^{-2}$ | ▼ |
| Stage: IV | - | - | | - | | - | | - | - | | $1.2\times10^{-2}$ | △ |
| ΔMMR*3 | - | $7.2\times10^{-7}$ | △ | - | | - | | - | - | | - | |
| No.Kataegis | - | $2.5\times10^{-2}$ | ▼ | $3.0\times10^{-5}$ | △ | $1.3\times10^{-2}$ | ▼ | - | $3.1\times10^{-2}$ | △ | - | |
| No.Omkili | $2.1\times10^{-2}$ △ | - | | $4.6\times10^{-2}$ | △ | - | | - | - | | - | |
| ΔSMG*4 | - | ACVR2A, PIK3CA, RNF43, ARID1A, ERBB2, TP53, B2M, MAP2K7, PTEN, MUC6, ARID2, KRAS | | TP53 | | TGFBR2, ELF3 | | - | - | | - | |

*WGD* Whole-genome doubling, *SCNA* Somatic copy number alterations, *MMR* Mismatch repair, *SMG* significantly mutated genes.
The upward and downward arrowheads appended to each *P*-value indicate the increments and declinations observed in the target group, respectively. *P* values were calculated using two-sided Welch's t-tests (No. SV, No.mutation, WGD, SCNA, Age, No. Kataegis, and No. Omikili) or two-sided Fisher exact tests (diffuse and intestinal type, smoking habit, stage, and mutation of MMR/SMG). Items that did not show significant differences (*P* > 0.05) were excluded.
*1: 24 cases of hypermutation were considered outliers and excluded from Welch's t-test.
*2: UICC tumor stage classification.
*3: ΔMMR refers to mutations in MLH-1/3, MSH-2/3/6, and PMS-1/2.
*4: SMG are defined by Totoki. et al. [15] from non-hypermutated cases of gastric cancer.

associated with mismatch repair deficiency and were consistently enriched with mismatch repair gene mutations ($P = 7.2 \times10^{-7}$, Table 1). This subtype included an RS4/RS2 mixed subgroup (Fig. 2b), showing a lower SV frequency ($P = 1.2 \times10^{-15}$) and enrichment of both microsatellite instability (MSI) and hypermutation cases ($P = 1.5 \times10^{-2}$ and $P = 6.9 \times10^{-4}$, respectively).

In contrast, cases in subtype RS6 had fewer point mutations ($P = 4.6 \times10^{-2}$) and clusters of SVs containing super-large SVs (> 10 Mb), and were associated with segment amplifications. *Kataegis* was positively correlated with RS6 (Supplementary Data 8) and was frequently observed in the subtype RS6 ($P = 3.1 \times10^{-2}$, Table 1). Molecular features of subtype RS2/6 were similar to those of subtype RS6; however, diffuse-type cases were characteristically enriched in subtype RS2/6 (19/38, $P = 7.7 \times10^{-4}$). Cases of subtypes RS6 and RS2/6 had poor prognoses (Fig. 2c). These two subtypes contained more advanced-stage cases ($P = 1.3 \times10^{-2}$ and $P = 3.0 \times10^{-2}$, respectively) (Table 1).

Correlations between the seven SV subtypes and four molecular classes, Epstein-Barr virus (EBV), MSI, genomically stable (GS), and chromosomal instability (CIN), reported in The Cancer Genome Atlas (TCGA) project[1] were examined. The subtype RS2 contained MSI- and EBV-positive cases (Supplementary Fig. 13a). The subtypes RS4 and RS2/6 had significantly more diffuse-type cases ($P = 1.7 \times10^{-2}$ and $7.7 \times10^{-4}$, respectively) with characteristic driver alterations (*CDH1* and *CLDN-ARHGAP* fusions) (Supplementary Fig. 13b), which were similar to TCGA GS class. The subtype RS3 had more intestinal-type cases ($P = 9.1 \times10^{-3}$) and copy number alterations ($P = 4.1 \times10^{-2}$) that corresponded to TCGA CIN class (Table 1).

## Molecular classification of SV clusters (SVCs) in GC

SVCs have been frequently detected in cancer genomes[7–16] and are associated with oncogene amplifications[17]. Approximately 36.6 % (18,118/49,509) of the total SVs were clustered and formed 3,457 SVCs in GC, and the ratio of SVCs varied among the RS subtypes. Less than 20% of SVs in subtype RS2 belonged to SVCs, whereas > 50% were clustered in subtypes RS3 and RS6 (Supplementary Fig. 14). We classified all 3,457 SVCs based on their profiles characterized by SV type, size, distribution, ΔCN, and SVAF (see **Methods**). We then annotated the proposed molecular mechanisms[29]: CFS, L1-retrotransposition, non-allelic homologous recombination-mediated duplication (NAHRD), and breakage-fusion-bridge cycles (BFBC) (Fig. 3a, Supplementary Data 9, and Supplementary Fig. 15).

CFS-like SVCs were characterized by recurrent deletions of various sizes within the same region and gradual attenuation of copy number (Fig. 3b). Among the 793 CFS-like SVCs, 38.9% (309/793) overlapped with validated CFS segments that were registered on HumCFS[30]. The SVCs in subtype RS4 (83.3%, 100/120) were dominated by the CFS-like type (Fig. 3a). The L1-retrotransposon-type SVCs were located within 1.0 kb of the LINE1 elements (Fig. 3c) and were frequent in subtype RS5 (24.9%, 50/201) (Fig. 3a).

Both NAHRDs and BFBCs cause gene amplifications[29]. NAHRD includes local gene amplifications triggered by medium-sized tandem duplication that generates adjacent low-copy repeats and induces template errors recurrently[31,32] (Fig. 3d). NAHRD-type SVCs were characterized by nested tandem duplications, followed by an increased copy number ratio (≥ 1.5) at the breakpoints (Supplementary Data 9). In our cohort, 269 SVCs were classified as the NAHRD type and included 15.4% (232/1503) of SVCs in subtype RS3. In BFBC-type SVCs, telomere end

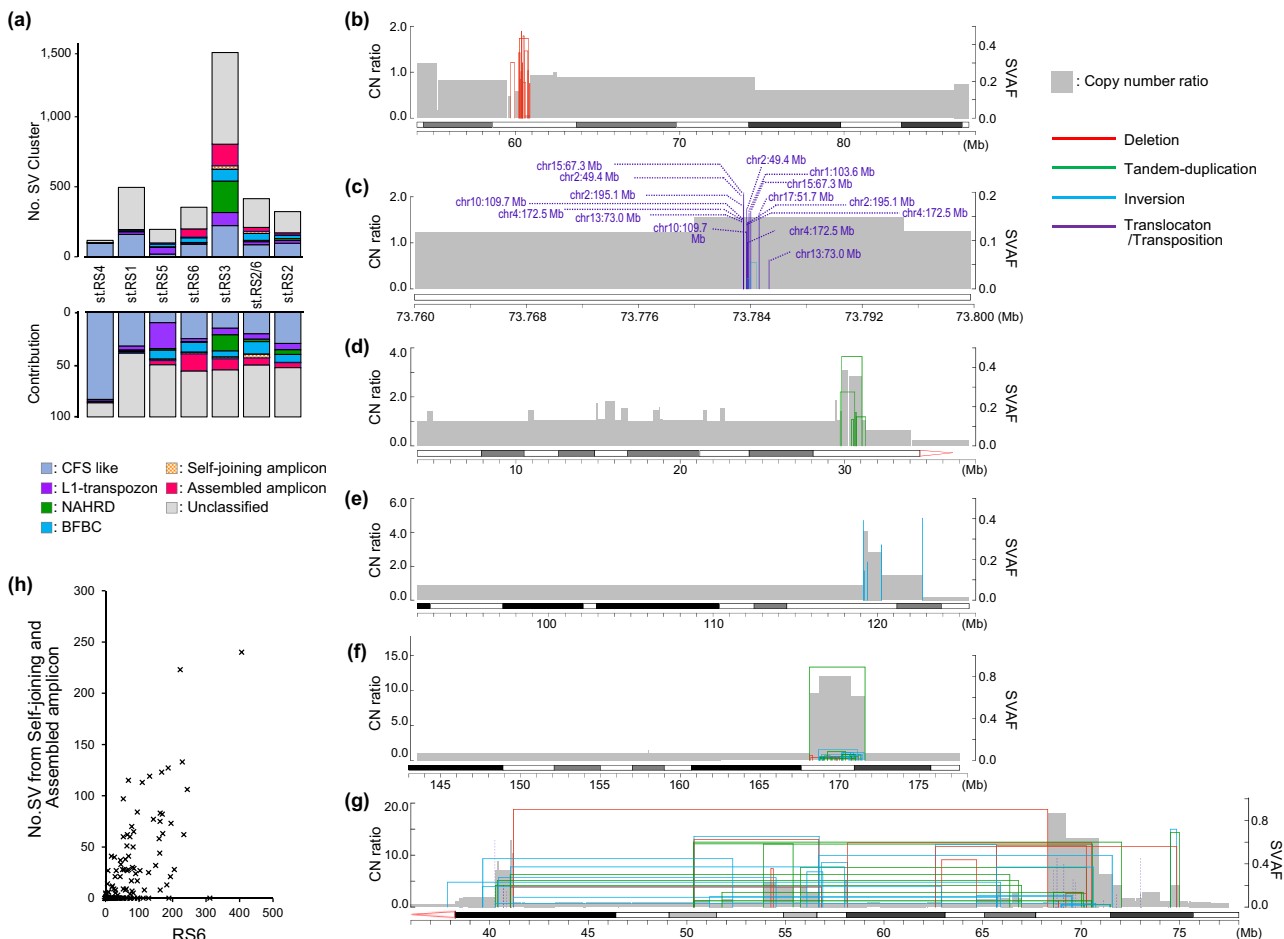

**Fig. 3 | Molecular classification of SV clusters (SVCs). a** Frequency of the six SVC types in each RS subtype. The numbers and contributions of each SVC type to the seven RS subtypes are shown. **b–g** Representative molecular presentations of SVC types. The X-axis corresponds to the chromosomal positions of SV breakpoints. The primary and secondary Y-axes indicate copy number ratio and adjusted structural variant allele frequency (SVAF), respectively. Colored lines indicate each SV type: deletions (red), tandem duplications (green), inversions (cyan), and translocations (purple). **b** A case of common fragile site fusion causes dicentric chromosomes, followed by rupture and fusion cycles[33]. The BFBC-type SVCs showed recurrent fold-back inversions with stepwise copy number increase (Fig. 3e) and were most frequently (11.9%) observed in subtype RS2/6 (Fig. 3a). Foldback inversions were significantly found in BFBC-type SVCs ($P < 10^{-7}$), while nested tandem duplications occurred more frequent in other types of amplified SVCs (Supplementary Data 10).

(CFS)-like SVC type at 3p14.2, *FHIT* locus (#pfg120T). **c** L1-retrotransposon SVC at 8q21.11, *HNF4G* locus (#GC440T). **d** Non-allelic homologous recombination-mediated duplication (NAHRD) SVC at 16p11.2 (#GC371T). **e** Breakage-fusion-bridge cycle (BFBC) SVC at 11q13.3, *CCND1* locus (#GC164T). **f** Self-joining amplicon SVC at 3q24-q26.32 (#GC416T). **g** Assembled amplicon SVC located at 12q12-q21.2 (#pfg116T). **h** Correlation between the number of RS6 and self-joining and assembled amplicon SVCs. **a–h** Source data are provided as a Source Data file.

## Extrachromosomal DNA-driving SVC generates driver oncogene amplicons in GC

Our analysis also identified other complex types of SVCs that were not classified, as described above. They exhibited high SVAF breakpoints at the amplicon edges and contained a group of lower SVAFs that were densely located within the amplicon (Fig. 3f). In some cases, these amplified segments were connected beyond > 2 Mb distance (Fig. 3g). We named the former a self-joining amplicon, and the latter an assembled amplicon. The number of these complex SVCs correlated to that of the "clustered large-scale SV" in the RS6 signature (r = 0.70, $P = 9.2 \times 10^{-26}$) (Fig. 3h).

Furthermore, multiple similarities were observed between ecDNA-associated SVs and self-joining/assembled amplicon SVCs

(Fig. 4a). First, the large-scale and high SVAF-SVs at the amplicon edges in the self-joining/assembled amplicon SVCs were consistent with the initial junctions of ecDNA (Fig. 4a, solid circle in RS6 and Step 1). Second, the clustered low SVAF-SVs corresponded to further SV accumulation in ecDNA, generated by error-prone replication during multiple rounds of mitosis (Fig. 4a, dashed circle on RS6 and Step 2a). Third, the ultra-large-scale SVs that connect intra-cluster SVs and distantly located genomic regions corresponded to the reintegration of ecDNA into the genome (Fig. 4a, Step 2b), which is a source of homogeneously stained chromosomal regions (HSRs)[34]. By applying the criteria described in the **Methods** to identify ecDNA-reintegrated SVs, 471 SVs were identified as potential reintegrations (Supplementary Data 11 and Supplementary Fig. 16a). Next, we performed fluorescent in situ hybridization (FISH) analysis of the four cases of them and validated all cases showed signals of both ecDNA and HSR (Fig. 4b and Supplementary Fig. 16b–d). *Kataegis* and *omikli* were enriched in the BFBC, self-joining, and assembled amplicon regions (Supplementary Fig. 17), especially high in the latter two types, consistent with a previous report showing *kataegis* accumulation in ecDNA[35].

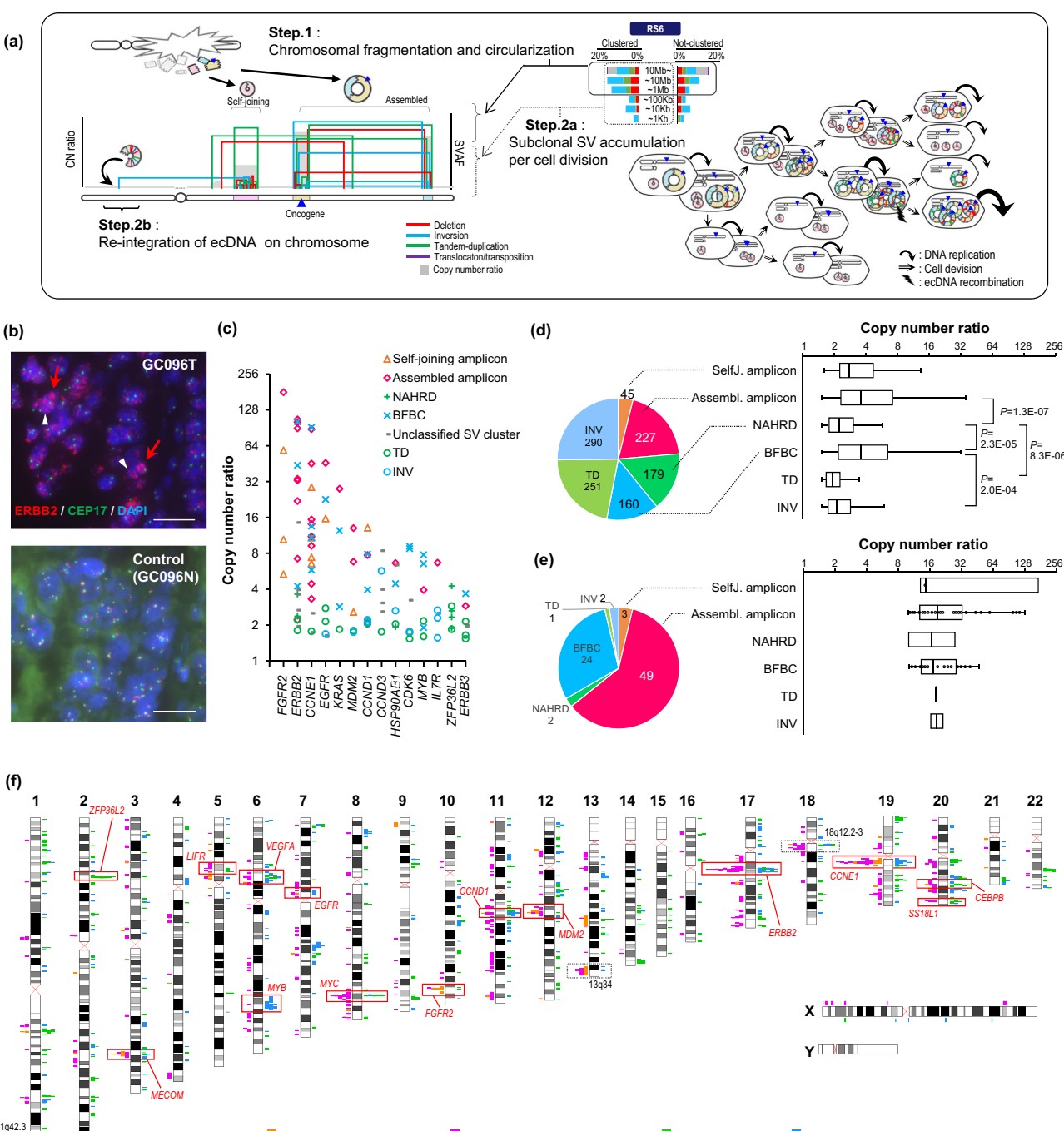

**Fig. 4 | Molecular characterization of amplicon-associated SVCs. a** Schematic diagram summarizing the consistencies of amplicons SVC and RS6 and comparison of their SV patterns and extrachromosomal DNA (ecDNA) dynamics. The left panel shows the breakpoint location, copy number ratio, and SVAF of a SVC. The figure illustrates the steps of ecDNA generation, which are expected to correspond to the amplicon SVC. Step 1: After large-scale chromosomal fragmentation, the exposed ends of DNA fragments are joined via DNA repair mechanisms, resulting in self-joined ecDNA (formed by a single fragment), or assembled ecDNA (by randomly connecting multiple fragments). Step 2a: During cell cycle, additional SVs (detected as lower SVAF-SVs) accumulate inside the ecDNA. An ecDNA including oncogene (shown as blue triangles) is further amplified, and more SVs are accumulated. Step 2b: In cases ecDNA re-integrates chromatin distant from its original coordinates, the breakpoints appear as an ultra-large SV. RS6 shows in simplified representation. **b** Representative *FISH* analysis validating extrachromosomal DNA (ecDNA) and homogeneous staining of the chromosomal region (HSR) at *ERBB2* gene locus in GC096T. Scattered spots indicate ecDNA (red arrow) and clustered bright stains indicate HSR (white arrow). Signals of *ERBB2*, the centromere of chromosome 17, and DAPI are shown in red, green, and blue, respectively. Scale bars, 20 μm. FISH were performed for two time, with similar results. **c** SVC types and copy ratio of amplified GC oncogenes in this cohort. **d** Distribution of amplification intensity for each type of SVC and SV (copy number ratio >1.5). The pie chart shows the proportion of each type and its sample size. In the box plots, the centerline, box limits, and whiskers indicate median, upper and lower quartiles, and 1.5× interquartile range, respectively. Each significance is tested by a paired two-sided Fisher's exact test. **e** The distribution of amplification intensity only with large-scale amplification (>10.0). **f** The ideogram represents the genomic landscape of the four SVC types accompanied by segmental amplifications. The colored boxes indicate the location of each type of amplicon. Putative GC oncogenes are shown by red-colored frames, and uncharacterized SVC hotspots are shown by dotted frames. **b**–**f**, Source data are provided as a Source Data file.

Most oncogene amplicons, such as *ERBB2*, *CCNE1*, and *EGFR*, were associated with a mixture of four SVC types, while some loci were characterized by their propensity for an SVC type: 2p21 (super-enhancer segment of *ZFP36L2*[23]) with the NAHRD-type, 6q23.3 (*MYB*) with the BFBC type, and 10q26.12-13 (*FGFR2*) with the self-joining amplicon type (Fig. 4c). Among these amplicon-associated SVC types, the assembled amplicon and BFBC types were associated with significantly higher copy numbers than the NAHRD type ($P = 1.3 \times 10^{-7}$ and $2.3 \times 10^{-5}$, respectively; Fig. 4d). The assembled amplicon type was the most frequent (37%) in all clustered SVs, followed by NAHRD (29%) and BFBC (26%) (Fig. 4d). The assembled amplicon type consisted of approximately two-thirds of the highly amplified genomic regions (copy number ratio > 10) (Fig. 4e), supporting its association with ecDNA, as previously reported[10].

The chromosomal distribution of these SVCs with amplicon types revealed that they were enriched in specific genomic regions that included GC oncogenes (Fig. 4f). Moreover, previously reported ecDNA-associated amplicon loci[10,17] overlapped with self-joining/assembled amplicon SVCs (Fig. 4f). In addition to GC oncogene loci, we discovered SVC-enriched amplified loci, such as 1q42.3, 13q34, and 18q12-13, where the target genes remain to be determined. Among these, 1q42.3 genomic fragment was reported to induce cellular senescence[36], implying the existence of an unexplored oncogene.

## Discussion

By analyzing 170 GC whole genomes, we searched for potential GC driver SVs that are recurrent and have a high SVAF. We focused on tandem duplication and inversion that cause segmental amplification and identified 27 hotspot loci, which were classified into singular and multiple types. The singular type is characterized by gene fusions, disruptions of tumor suppressor genes, and low-level duplications of gene regulatory elements. In contrast, multiple types were generated by a mixture of different SV types, resulting in higher copy number amplifications. SV numbers varied significantly and occasionally accompanied the complex SVCs in these cases. These SVCs are characterized by accumulated and sub-clonal SVs, which are progressively amplified by inaccurate double-strand DNA break repair and chromosome segregation. These features represent specific distribution patterns for driver SVs.

Studies across human cancer types have identified multiple RS[8,37]. The present study extracted six RS in GC, and hierarchical clustering identified seven GC subtypes in which one or a few RS were dominant. This biased distribution suggests that unknown external (epidemiological background) and/or internal (somatic molecular alterations) factors are involved in RS generation. Consistently, subtype RS1 was associated with defective homologous recombination-based repair (BRCA), subtype RS2 with mismatch repair deficiency, and subtype RS3 with *TP53* mutations and smoking habits. Although smoking is a validated risk factor for GC, the smoking-associated mutation signature was not frequent in GC. Our results disclose that a specific environmental factor is associated with somatic rearrangement patterns. Characteristic chromosomal instabilities were also annotated to specific RS, such as NAHRD (RS3), L1-retrotransposon (RS5), and CFS (RS4).

RS6 has also been identified in breast and liver cancers[8,37] and oesophageal adenocarcinomas[18]. Although its nature remains unclear, it is enriched with complex SVCs. By molecular dissection of these SVCs, we discovered two amplicon-generating SVC types, self-joining and assembled amplicons, significantly correlated with RS6. These amplicons shared similarities with ecDNA-associated SVs, which were further verified using FISH analysis. Our study linked the characteristic SVAF pattern of clustered SVs to the process of ecDNA expansion and uncovered that ecDNA is significantly involved in *FGFR2* and other GC oncogene amplifications, preferentially occurring in a subset of poorly prognostic GC cases.

## Methods

### Gastric cancer (GC) study cohort

All procedures performed in studies involving human participants were in accordance with the ethical standards of the institutional and/or national research committee and with the 1964 Helsinki Declaration and its later amendments or comparable ethical standards. This study has been approved by the Institutional Review Board of the National Cancer Center Japan (approval number. G20-03). Written informed consent was obtained from all participants in the study. The information on gender and/or sex, number, and age of all participants in this study is presented in Supplementary Data 4.

### Whole genome sequencing (WGS)

Genomic DNA was extracted from the tumor and matched normal tissues. Libraries with an insert size of 350–550 bp were prepared from 2 μg of sonicated DNA using a TruSeq DNA PCR-free kit (Illumina, San Diego, CA, USA) according to the manufacturer's protocol. The libraries were sequenced using a HiSeq 2500 instrument (Illumina) with PE reads of 101 bp. Sequencing reads were aligned to the human genome reference assembly GRCh37 (hg19) using Burrows-Wheeler Aligner- Maximal Exact Match (BWA-MEM)[38].

### Mutation calling

Paired-end reads were aligned to the human reference genome (GRCh37) using NovoAlign (http://www.novocraft.com/products/novoalign/) for both tumor and non-tumor samples. Potential PCR duplications in which PE reads aligned to the same genomic position were removed, and pileup files were generated using SAMtools[39] and in-house programs. A mapping quality score of at least 20 and a base quality score of at least 10 were used as the cut-off values for base selection. Somatic mutations were selected using two filtering conditions: 1) the number of reads supporting a mutation in each tumor sample was at least four, and at least one base quality score at the mutation position of these reads was > 30; 2) the VAF of the matched non-tumor sample was <0.03, with a read depth of at least eight. To further exclude sequence context-dependent errors, the sequence reads of all non-tumor samples were grouped and used to discriminate true positives from false positives. NVAF, a VAF in grouped non-tumor samples with a sequence depth $\geq 10$ and VAF < 0.2, was calculated at each mutated genomic position. The following filters were applied: NVAF < 0.03 or 0.01 for TVAF $\geq 0.15$ or 0.15 > TVAF $\geq 0.05$, respectively, and the ratio of TVAF to NVAF was $\geq 20$. At each mutated genomic position, (vii) the ratio of non-tumor samples with a VAF $\geq 0.1 < 0.002$. Finally, mutations with a strand bias (between forward and reverse reads) of > 95% were removed. Single-nucleotide mutations were extracted from all these somatic mutations and used as input of SigProfilerClusters[40] to identify the *omikli* and *kataegis* region in our 170 WGS data.

### Mutational signature analysis

The contribution of the SBS signatures was calculated using the deconstructSigs[3], R package with the COSMIC signature datasets (https://cancer.sanger.ac.uk/signatures/signatures_v2/, https://cancer.sanger.ac.uk/signatures/dbs/, and https://cancer.sanger.ac.uk/signatures/id/). The clustering of GC cases by the contribution of mutational signatures was performed using unsupervised hierarchical clustering with cosine distance and Ward linkage. Kaplan–Meier survival curve analysis followed by a log-rank test was performed to estimate and compare the survival of the two groups (SBS17 > 50% vs. SBS17 $\leq 50$%). Epigenomic segments were identified using the segments defined in the human adult stomach mucosa using the RoadMap Epigenomics Consortium[41]. The chromatin regions with one of the states from E1 to E8 and their regulatory elements were considered 'active' regions, and the remaining regions were designated 'inactive'. The total size of the active and inactive

chromatin regions, excluding the unmappable regions, were 486,852,497 bp and 2,374,480,109 bp (17:83), respectively. The contribution of the signature in chromatin-active and -inactive areas was assessed using Fisher's exact test.

## Copy number analysis

The initial copy number ratio was estimated by comparing the read depth of the tumor and normal samples per bin (5,000 bp). To segment the copy number ratio, we used the DNAcopy Bio-Conductor package[42]. The segmented $\log_2$ copy number ratio was adjusted based on the tumor purity of each sample using the following formula;

$$R'(x) = \frac{1 - (1 - R(x))}{\text{tumor purity}} \tag{1}$$

where R′(x) is the adjusted copy number ratio and R(x) is 2^(segmented log2 copy number ratio).

## Whole genome doubling

We performed FACETS (v.0.6.0)[43] to determine allele-specific copy numbers.

The following parameters were used: snp.nbhd = 500, pre-ProcSample's cval = 50, and procSample's cval = 300. We inferred whole-genome duplication, in the same method as previously reported[44], when the frequency on the genome with a major copy number of 2 or more was greater than 0.5.

## Structural variant calling

We developed an in-house pipeline (callallSV) to detect all types of SVs including deletions, tandem duplications, inversions, and translocations. Paired-end reads of 100 or 150 bp of approximately 350–550 bp fragments from paired tumor and non-tumor samples were used as inputs to the pipeline. All PE reads were aligned to the human reference genome build hg19 using BWA-MEM[38] with the -T 0 option. After removing PCR duplicates, reads with low alignment quality were ruled out if the mismatched bases were > 20% of the read length or if no end reads were uniquely aligned. After filtering, the datasets were processed using two independent algorithms (Supplementary Fig. 2). The first employed PE reads mapped discordantly, for which both ends aligned to the reference genome uniquely with improper spacing, orientation, or both[45]. After excluding uncertain alignments (mapping quality score <37, or the number of mismatches > 2), these discordant reads were clustered based on their orientation and pair-mate locations. Rearrangements were then identified using the following analytical conditions: (i) forward clusters and reverse clusters were constructed from the end sequences aligned with forward and reverse directions respectively; (ii) two reads were allocated to the same cluster if their end positions were not farther apart than the maximum insert distance of pair end library; (iii) clusters with a distance between the leftmost and rightmost reads that was greater than the maximum insert distance were discarded; (iv) paired end reads were selected if one end sequence fell within the forward cluster and the other end fell within the reverse cluster (we hereafter called this pair of forward and reverse clusters as paired clusters); (v) if paired clusters overlapped with other paired clusters, all of the overlapping paired clusters were discarded; (vi) for the tumor genome, rearrangements predicted from paired clusters which included at least four pair s of end reads and at least one pair of end read s perfectly matched to the human reference genome, were selected; (vii) for the non tumor genome, rearrangements predicted by at least one pair of end reads were selected. By comparing the predicted rearrangements in the tumor and non-tumor genomes, somatic rearrangements that were only detected in the tumor genome were identified. The second method

used single reads that were split and mapped apart (so-called "soft-clipped reads") to identify SV breakpoints. All soft-clipped reads were extracted as SV candidates if they satisfied the following conditions: alignment score > 20, number of mismatches <10, the difference in score between the best alignment and the second one > 1, and no breakpoint observed from control samples (allowing one read or <1% of the detected reads in the tumor sample). The sequences of the upstream and downstream regions of each breakpoint were reconstructed to obtain a rearrangement sequence. All input reads were realigned to the reconstructed sequences, and the reads that were better mapped to the reconstructed sequence were counted as SV-supporting reads. The remaining reads aligned better with the reference genome were counted as reference-support reads. The numbers of SV- and reference-support reads from tumor and non-tumor samples were evaluated using Fisher's exact test.

Finally, the outputs from the two algorithms, using PE and soft-clipped reads, were integrated. False-positive SVs were filtered out based on the following cut-off values: (i) for translocation SVs, the number of support reads was ≥ 8 with ≥ 4 PE reads, and for other SV types, it was ≥ 4 with ≥ 2 PE reads; (ii) read depth at SV breakpoint was ≥ 10; (iii) SV allele frequency was ≥ 0.07; (iv) the total length of the alignment region of soft-clipped reads supporting an SV was at least 1.6 times the read length. Using with these conditions, 49,059 SV (22,179 deletions, 8534 inversions, 11,234 tandem duplications, and 7112 translocations) were obtained.

The callallSV software is freely available for non-commercial use at https://github.com/ma9606/callallSV. All researchers and technicians involved in academic genomic analysis can re-use or adapt the callallSV code (released under the GPL-3.0 license) to implement similar tasks.

## PCR validation

PCR validation of the predicted somatic SVs was performed to determine the appropriate parameters in the SV caller pipeline. We randomly selected 123 SVs (53 deletions, 18 inversions, 36 tandem duplications, and 16 translocations) for validation analysis (Supplementary Data 12). These validated SVs were selected from four SV types with variations in their support read depth, support type (PE or soft-clip), and total length of the alignment region (see also "Structural variant calling"). The DNA fragments of the tumor genome containing the breakpoints of 123 SVs were amplified, and the exact breakpoints of 100 SVs were determined using Sanger DNA sequencing. A total of 100 SVs were validated as somatic events by comparing the corresponding non-tumor genomes. Of the remaining twenty-three, 15 could not be amplified or sequenced because of surrounding repetitive sequences, and eight could not be validated. Therefore, the prediction accuracy of our approach for detecting somatic SVs was 92.6% (100/108).

## Benchmark of in-house SV caller (callallSV)

To evaluate the reliability of SV detection, we validated callallSV using 16 cases as the test set. The callallSV detected 3898 SVs, and 2634 SVs were detected using GenomonSV[46,47] (version 2.5.0, https://github.com/Genomon-Project/GenomonSV) in the default setting as the benchmark set. Of these SVs, 2384 were detected using both tools, while 1514 SVs were uniquely detected using callallSV. We randomly selected 56 SVs from these 1514 unique SVs detected using callallSV and validated them using RT-PCR and Sanger DNA sequencing; 89.3% (50/56) of the SVs were verified. For each number of reads supporting an SV, we calculated the true positive rate (TPR or sensitivity) at which our callallSV detected benchmark SVs and the false positive rate (FPR) at which our callallSV detected non-benchmark SVs. Plotting the ROC curve using these values resulted in an area under the curve of 0.766 (Supplementary Fig. 18).

## Estimation of SVAF

We adjusted the SVAF for tumor sample $i$ and genomic position $x$ by tumor purity as follows:

$$\text{vaf}(x, i) = \frac{\text{Rsv}(x,i)}{\text{Rsv}(x,i) + \text{Rref}(x,i)} \quad (2)$$

$$\text{SVAF}(x, i) = \frac{\text{vaf}(x,i)}{\text{purity}(i)} \quad (3)$$

where Rref is the number of reads supporting the reference genome, and Rsv is the number of reads supporting the SV. The copy number ratio was also adjusted by tumor purity and used to judge the loss of heterogeneity (LOH) or amplification.

$$\text{acn}(x,i) = 1 + \frac{(\text{copy number ratio}(x, i) - 1)}{\text{tumor purity}(i)} \quad (4)$$

When two breakpoints from an SV had different $\text{acn}(x, i)$ values, the breakpoint closer to 1 was used. If $\text{acn}(x, i)$ was <0.75, the position was considered to be LOH, and the following formula was applied:

$$\text{SVAF}(x, i) = \left(\frac{\text{vaf}(x,i)}{\text{tumor purity}(i)}\right) * (2 - \text{tumor purity}(i)) * 0.5 \quad (5)$$

If $\text{acn}(x, i)$ was >1.25, and the ratio of vaf to tumor purity was ≥0.60, the position was considered amplified, and the SVAF was adjusted as follows:

$$\text{SVAF}(x, i) = \frac{\text{vaf}(x,i)}{\text{tumor purity}(i)} \times \frac{\text{acn}(x,i)}{(\text{acn}(x,i) - 0.5)} \times 0.5 \quad (6)$$

The SVAFs of all SVs were calculated except for those not supported by sufficient reads (≥10) to reduce misestimation. A correlation plot of copy number and SVAF showed that the function appropriately adjusted for the effect of copy number alteration (Supplementary Fig. 19).

## Whole transcriptome RNA sequencing (RNA-seq) analysis

We prepared 150–200-bp insert libraries from total RNA using the SureSelect Strand Specific RNA Library Prep Kit (Agilent Technologies) together with the TruSeq stranded mRNA Sample Prep Kit or the TruSeq mRNA-Seq sample preparation kit (Illumina). The libraries were sequenced using 100-bp PE sequencing on HiSeq 2500, HiSeq2000, or GAIIx (Illumina) according to the manufacturer's instructions.

The gene fusions were identified using our in-house pipeline[48] and fusion (https://github.com/Genomon-Project/fusion), enabling a valid selection of putative chimeric transcripts generated by the STAR[49] algorithm, as previously reported[50].

RNA-seq reads were aligned to the human transcriptome (UCSC gene) and genome (GRCh37/hg19) references using BWA to calculate gene expression. After the transcript coordinates were converted to genomic positions, an optimal mapping result was selected from either the transcript or genome mapping by comparing the minimal edit distance to the reference. Local realignment was performed using an in-house short-read aligner with a smaller k-mer size (k = 11). Finally, fragments per kilobase of exon per million fragments mapped (FPKM) values were calculated for each UCSC gene while considering strand-specific information.

## Annotation for regulatory sequences

To annotate the regulatory region in the GC genome, published H3K27ac peak data were downloaded from the Gene Expression Omnibus with accession number GSE51776[51]. Super-enhancers were identified using the rank-ordering of the super-enhancers algorithm with default parameters[52].

## Rearrangement signatures and subtype identification

Rearrangement signatures (RSs) were processed using a statistical framework[8], which was used to extract SBS signatures. First, the genomic region of the clustered SVs for an individual sample was identified as those with an average rearrangement distance at least 10-fold greater than the whole-genome average for that sample[8]. If a breakpoint of an SV was located on an SV-clustered region, it was classified as "clustered"; else, it was classified as "not-clustered." SVs were subclassified based on their type: deletions, inversions, tandem duplications, and translocations. The former three SV types were sub-classified according to the size of the SV, which was defined as the distance between two breakpoints of an SV ( < 1 kb, 1–10 kb, 10–100 kb, 100 kb–1 Mb, 1–10 Mb, and >10 Mb). The last SV type, translocation generated by crossing two chromosomes, was classified into retro-transposition and translocation classes by considering their break-point coordinates locations near LINE-1 transposable elements (<1.0 kb). All SVs were further subclassified according to the chromatin status (active or inactive) of the breakpoint genomic area. The chromatin status was defined according to the SBS signature analysis described above.

The detected SVs in 167 cases, excluding two cases where the number of detected SVs was <10, were represented as a matrix of 80 distinct categories and decomposed using nonnegative matrix factorization[53] (iterations = 100) in MATLAB (version 6.1.0.604, The MathWorks, Inc., USA). Rearrangement signatures were extracted from each of the 167 cases, and seven subtypes were identified by consensus clustering[54] with a hierarchical clustering algorithm using cosine similarity distance with Ward2 linkage.

## SVC profile

SVCs were classified based on the "SVC profile," which comprised the following factors (Supplementary Data 9 and Supplementary Fig. 15):

- Composition of SV type
- Average copy number ratio ($\overline{\text{CNr}}$)
- Maximum difference in CNr at the SV breakpoint (ΔCNr)
- Stepwise copy number change at the SV breakpoint
- SVAF, breakpoint coordinates, size of each SV constituent of the SVC

Based on these profiles, SVCs were first classified using known molecular mechanisms as follows:

CFS like: More than half of SVs are occupied by deletions and have stepwise copy number reductions caused by recurrent deletions[55]. One or more deletions decreased the copy number ratio at breakpoints of >0.2.

L1-transposition: More than half of the SVs are occupied by translocations whose breakpoints are concentrated in a narrow range (<500 bp) and located within 0.1 kb from the nearest LINE1 transpo-sable element[56,57], as annotated with RepeatMasker 4.0.9[58].

NAHRD: More than 40% of SVs are occupied by tandem dupli-cation, and at least half overlap in their spacer, defined as the region between breakpoints of a tandem duplication[31]. Here, 90% of the spacer length was used as the threshold for SV overlap, and this type of SVC contains one or more nested pairs of tandem duplications, i.e., a spacer with a high SVAF-SV including a lower SVAF-SV. Fur-thermore, it overlaps with the "amplified segment," where the copy number ratio change at its boundary is >1.5, and the copy number average is >4.0.

BFBC: More than 40% of SVs are occupied by inversions, and at least half of these should be fold-back inversions, a trace of BFBC. This SVC also meets the following three criteria[33]:1) the absence of reci-procal partner of an inversion; 2) the inversion-induced copy number change (q < 10 × 10⁻³) with at least one side of its breakpoint; and 3)

the distance between the two ends of the inversion is <30 kb. This type also overlaps with the amplified segments, as described above.

A total of 1957 SVCs did not match the above classification, and 56 and 274 SVCs showed characteristic properties and were annotated as "self-joining amplicon" and "assembled amplicon," respectively. These undefined types of SVCs overlap the above-mentioned amplified segments and are defined as follows:

Self-joining amplicon: The breakpoint(s) of a high SVAF-SV, which ranked within the top 30% of SVAF in the SVC, demarcated the SVC region and matched the boundary of an amplified segment; more than half of the SVs are located between the high SVAF breakpoints.

Assembled amplicon: A high SVAF-SV in the top 30% SVAF of the SVC connects the boundaries of two or more independent amplified segments separated by over 2 Mb on the chromosome; lower SVAF-SVs ($\geq$ 1) connect the segments or are located within one segment.

In cases where these two types SVCs contain an SV that meets the following three criteria, the SV was considered as a sign of chromosomal reintegration: (1) one of the two breakpoints from an SV located in a self-joining amplicon or an assembled amplicon, and the other is located in a non-SV clustered region; (2) the Breakpoint of a non-SV clustered region is not accompanied by a copy number change (dCN is less than 0.5) before and after its coordinate; (3) the size of an SV spans more than 2 Mb, or its type is translocation-SV type.

## Cell culture

Human GC cell lines HGC27 (RCB0500) and MKN45 (RCB1001) were purchased from the Riken Cell Bank (Tokyo, Japan) and cultured in Dulbecco's modified Eagle's medium (DMEM) with 10% FBS (Sigma-Aldrich, St. Louis, MO, USA).

## Fluorescent in situ hybridization (FISH)

FISH analysis was performed on 4-µm-thick formalin-fixed paraffin-embedded (FFPE) sections. Custom *FGFR2* break-apart FISH assays were performed using a probe set that hybridized with the neighboring 5′-telomeric (RP11-78A18, labeled with Spectrum Green) and 3′-centromeric (RP11-7P17, labeled with Spectrum Red) sequences of *FGFR2* (Chromosome Science Labo Inc., Sapporo, Japan). *EGFR* and *ERBB2* were analyzed using Vysis EGFR/CEP7 FISH probe kit (Abbott, Illinois, USA) and PathVysion HER2/CEP17 FISH probe kit V2 (Abbott), respectively. One hundred non-overlapping cells with FISH signals were examined, and a detailed signal pattern was recorded at a clinical laboratory (LSI Medience Corp., Tokyo, Japan).

## siRNA transfection and cell growth assays

siRNA oligonucleotides targeting human *BCAR4* (R-030589), *CEBPB* (L-006423), and a non-targeting negative control siRNA (D-001810) were synthesized by Dharmacon (On-TargetPlus SMART pool; Lafayette, CO, USA). Cells were seeded into 96-well plates (1,500 or 2,000 cells per well) in an antibiotic-free culture medium. The next day, mixtures of siRNA and Lipofectamine RNAiMAX reagent (Thermo Fisher, Waltham, MA, USA) were added to each well, along with 9 pM siRNA solutions. After 24, 48, and 72 h of transfection, viable cell numbers were counted by the MTT assay using Cell Count Reagent SF (Nacalai, Tokyo, Japan). Three days after transfection, total RNA was extracted and reverse-transcribed to cDNA using a FastLane Cell cDNA kit (Qiagen, Venlo, Netherlands). RT-qPCR was performed using KAPA SYBR Fast qPCR Master Mix (KAPA Biosystems, Wilmington, MA, USA) and *BCAR4*, *CEBPB*, and *GAPDH* specific primers on a Light Cycler 96 platform (Roche, Basel, Switzerland). The values obtained by RT-qPCR were normalized to those for *GAPDH*. The sequences of the siRNAs and RT-qPCR primers are listed in Supplementary Data 13.

## Reporting summary

Further information on research design is available in the Nature Portfolio Reporting Summary linked to this article.

## Data availability

All data needed to evaluate the conclusions in this study are presented in this paper, the Supplementary Materials, or are available at the following repository. The WGS data of 81 Japanese cases with their prefix 'GC' in this paper have been deposited in European Genome-phenome Archive (https://ega-archive.org/) with the accession numbers EGAD00001008610 and EGAS00001006051. Requests for academic purposes only will be processed by ICGC Data Access Compliance Office (https://docs.icgc-argo.org/docs/data-access/daco/applying) within ten business days. After access has been granted, the data is available for two years. The RNA sequencing data of the 62 Japanese cases generated in previous study[4] are available in the Japanese Genotype-phenotype Archive with the accession codes JGAS000228 and JGAS000229. For use, approval is required at the review by the NBDC Human Data Review Board. Data users shall apply for data use in accordance with the data use application procedures (https://humandbs.biosciencedbc.jp/en/data-use). These data are under controlled access because they are personally identifiable data defined by Japan's Personal Information Protection Law. The WGS data of 89 Chinese cases generated in previous study[19] with their prefix 'pfg' have been deposited in EGAD00001000782 and EGAS00001000597. The datasets are available under controlled access and access can be obtained by contacting the University of Hong Kong Gastric Cancer Genomics Study Data Access Committee. The remaining data are available within the Article, Supplementary Information or Source Data file. Source data are provided with this paper.

## Code availability

The callallSV software is freely available for non-commercial use at https://github.com/ma9606/callallSV under the GPL-3.0 license. All scripts used in the analyses presented here are also provided.

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

## Acknowledgements

We thank all patients who participated in this study. We also thank T. Taniguchi for sample collection and pathological advice, A. Matsumoto, A. Hara, T. Chikuta and E. Arakawa for expert technical assistance, and S. Yamamoto, S. Yamasaki, H. Ono, and E. Furukawa for data analysis. Computations were performed on the SHIROKANE supercomputer at the Institute of Medical Science, The University of Tokyo. This work

was supported by grants from the Practical Research Project for Innovative Cancer Control from the Japan Agency for Medical Research and Development (AMED) (21ck0106547h0002 to T.S. and 23jk0210009 to SY. and T.S); Research Grant of the Princess Takamatsu Cancer Research Fund (11-24308 to Y.T.), the National Cancer Center Research and Development Fund (2020-A-7 to T.S.), and JST CREST (M.K.).

## Author contributions

T.S. designed the study. H.R. contributed to sample acquisition and pathological evaluation. Y.A., F.H., and A.F. managed library preparation and sequencing. MSA, N. Hama, H. Nakamura, and Y.T. analyzed data. MSA, Y.T, Y.A., N. Hama, H. Nakamura, S.Y., MK, and T.S. interpreted data. MSA, N. Hama, Y.A., H. Nakamura, Y.T., and T.S. wrote the manuscript. All authors critically reviewed the manuscript.

## Competing interests

The authors declare no competing interests.
