## [Peer review file · Nature Communications]

REVIEWER COMMENTS

Reviewer #1 (Remarks to the Author): Expert in computational cancer genomics and structural variants

----- Overall evaluation -----

The presented manuscript focuses on the study of 170 whole-genome gastric cancers (GC). The authors analyze GC structural variants (SVs) and found patterns that lead to the identification of 6 rearrangement signatures (RSs). By investigating the impact of VSs, the authors identified 27 SV-hotspots as novel candidate GC drivers.

The study further analyzes the profiles of SVs taking into account SV length and allele frequencies, to provide insight into their contribution to GC affected oncogenes and their involvement in extra-chromosomal DNA linked to specific RSs. The manuscript provides valuable new data of whole-genome GC. It creates interesting hypotheses about the active SVs in GC and the mechanism of SV oncogene activation in GC. However, this reviewer has some comments that the authors must address to improve the quality and the rigor of the manuscript.

----- Major comments -----

1. In the section 'Recurrent SV hotspots represent new GC driver candidates', attempting to detect identify GC-driver SVs the authors state that they focused the analysis on two SV types (Inversion and tandem-duplication) and based on 4 ad hoc conditions they identify 27 SV hotspots region as driver candidates. However, in Supplementary Fig. 5 the authors show examples of results from translocation and deletion in know oncogenes. This appears a major incongruence in the result based on the authors' claim that the analysis was focused on Inversion and tandem-duplication. Moreover, this reviewer noticed that the SV calling found 22179 (45%) deletions and 7112 (14%) translocations. The prediction of the genomic region of SV hotspots candidate driver should be systematically presented by accounting for each SV type. The identification of a region of SV hotspots as driver candidates with the exclusion of highly prevalent SVs type, or the emphasis on specific SVs type warrants further in-depth explanations. They also can perform statistical analyses to provide the significance of these regions as the driver. It will be informative if the manuscript gives complete information about the DEL and TRA for each hotspot region in table 5.

2. Gene fusion is of the utmost importance in the cancer genome. Conversely, the manuscript scatters the mentions of genes fusion seemingly as supportive examples. The study implements a new SV caller and there is the availability of RNA-seq therefore the study should provide in the result a particularly comprehensive description of their gene fusion finding no limited but including the SVAF, the breakpoint location (specific exon, intron, or promoter).

3. They present a new pattern for SV signature showing its value on the analysis but, having a strong reference of SV signatures resulting from the recent publication of PCAWG project (Li, Yilong et al. "Patterns of somatic structural variation in human cancer genomes." Nature, 2020), they should establish an in-depth comparison pointing to the strength and weakness of their SV signature proposal.

4. In the section 'Extrachromosomal DNA-driving SVC generates driver oncogene amplicons in GC', the authors claim that SVCs with amplicon types are enriched in GC oncogenes, suggesting that they were positively selected for GC development. That statement must be supported by the corresponding statistical analyses, which should also include the confirmation that the enrichment is not due to SV signature thus, indeed it is a sign of their potential as positively selected regions.
5. In section 'Rearrangement signatures and subtypes in GC', The authors state that "RS4, consisting of large (0.1 to 1.0 Mb) deletions occurred in the active chromatin area" but Figure 2 does not support that statement. There is the same level of RS4 deletion in the inactive chromatin area.
6. The study implements a new method for detecting SVs from WGS without providing any benchmark with the established methods. The manuscript needs a proper benchmarking of the new method providing curves to interpret sensitivity, specificity, precision, and accuracy.
7. The section 'PCR validation' shows incongruent numbers. Initially, the authors state the validation of 120 predicted somatic SVs. But they precede by saying that they randomly selected 87 SVs. Therefore, it seems like the first statement is incorrect. Moreover, they continue by reporting the confirmation of 72 SV breakpoints out of the selected 87 but, it appears that they report a prediction accuracy of 96% (72/75), which is incorrect. What we can say from that data is 'in 72.5% (87/120) of the predicted somatic SVs, the sensitivity or true positive rate (TPR=TP/P) is 82.8 (72/87)'. This reviewer suggests revisiting the concepts in this section to provide the correct analysis by checking the validation of positive and negative predicted SVs.

----- Minor comments -----

1. Figure 1b does not show clearly the 27 SV hotspot regions identified as driver candidates.
2. Line 457, there is a repeated word: 'fusion'.
3. Supplementary Fig. 9, typo (4 instead of 1) in RS1.
4. The manuscript states, "The subtype RS2 contained MSI-positive and EBV-positive cases (Supplementary Fig. 10a)" but, based on the legend of Supplementary Fig. 10, this graph does not show MSI-positive cases.

Reviewer #2 (Remarks to the Author): Expert in gastrointestinal cancer genomics

Saito-Adachi et al provided an in-depth analysis of structural variations (SVs) in gastric cancer genomes and associated clinical features of the disease to molecular features identified through whole genome sequencing. This study provided a good overview of SVs and addressed several complex problems from identifying molecular processes generating SVs to their effect on driver genes with some relevant clinical

correlations. However there are several areas that requires clarification and streamlining to help present a clearer picture regarding the landscape of SVs in gastric cancer.

1. The authors used a new computational method to identify SVs in the paper that relies on the VAF of SV (SVAF). It is unclear how the VAF of amplified regions can be estimated robustly especially in cases of amplicons and ecDNA. An example in the supplementary data would illustrate the robustness of using SV VAF in classifying regions with SVs and including highly amplified regions. A correlation plot similar to Fig1a bottom with SVAF and CN as x and y axes would illustrate the relationship as well.

2. The section of SV signatures is very informative as the correlations with other clinical and molecular features provides insight on the molecular subtypes in GC. The addition of chromatin features provided an additional dimension to understand the processes generating the SVs. There are several minor comments on the methodology and presentation of the data. For the SV signature extraction, how many iterations were carried out and were the signatures presented in the manuscript robust? Similarly, the clustering of patients into subtypes can be assessed using consensus clustering to see how stable the clusters are.

3. The observation of a correlation between RS1 and the BRCA phenotype is clinically important. The addition of SNV and SV signatures provides a reliable way to identify GC patients with the BRCA phenotype with confidence especially with the addition of the inactive chromatin features. What does the SV signatures look like without the addition of the inactive chromatin features and how would the patients with the BRCA phenotype be grouped otherwise in other subgroups? Also, could the authors comment on how many cases would be missed if only the SV analysis is carried out, without the addition of the chromatin features?

4. It is important to match extracted SV signatures to reference signatures for the better comparison of signatures and prevent mis-identification of signatures. What are the signatures identified in this study and how do they relate to SV signatures in <https://signal.mutationalsignatures.com/> ? Are there new SV signatures that are previously not discovered?

5. The NAHRD signature closely resembles another tandem duplication signature known to be associated with CCNE1 amplifications (RS1 in signal). An analysis of tandem duplication sizes would be of interest to identify cases with small tandem duplications that might be missed in the signature analysis.

6. The grouping of patients according to SV signatures shows how diverse GC is as a disease and it would be useful to groups or subtypes better and with more informative names. St.RS2, STRS6 and STRS2/6 seems very distinct classes given the differences in SNV mutation counts and the presence of a MSI cluster within RS2. Can the MSI patients be split out as a cluster? Does stRS6 and stRS2/6 have similar

features and are the clusters separate clusters meaningful for interpretation? Clustering based on Ward.D2 might give a different clustering outcome.

7. The classification of SVs into classes according to the biological mechanism is challenging. This is especially in complex rearrangements where a large number of SVs are within a genomic region, where it is more challenging to identify each biological mechanism. It is unclear how the self-joining amplicons and assembled amplicons are different from ecDNA and if the additional categories provide additional information on the differences of the processes generating these SVs. Are there differences in the proportions of inversions, breakage-fusion bridges or other features in these complex SV classes that might point to different mechanisms?

8. Figure 3G needs to be refined as the lines are obscuring the copy number profiles. Authors can change the transparency value of the lines to show the profile better.

9. ecDNA and HSRs are difficult to differentiate without FISH staining done experimentally. Have the authors attempted to identify other cases with reintegration and if the SVAF provides a way to identify such cases?

Reviewer #3 (Remarks to the Author): Expert in gastric cancer genomics

Major Concerns:

1. Authors totally analyzed 170 gastric cancer (GC) genome sequencing WGS data, identified out 49,059 SVs. According to the recent pan-cancer analysis of whole genomes research, I was wondering about that weather the authors took diffuse hypermutation (omikli) and longer events (kataegis) signatures into concern.

2. On the other hand, is there any omikli or kataegis signature in the ecDNA author identified?

3. What's ration of the ultra-large-scale SVs that connect intra-cluster SVs and distantly located genomic regions corresponded to the reintegration of ecDNA into the genome in the cohort? not showed in figures.

4. Somatic mutations filters were based on RNA-seq, for low-frequency variants may not be adequately observed in genomic DNA but enjoy high read support in RNA if the corresponding genes are highly expressed. So, is there any additional somatic variant-calling pipelines can be contributed to the analyses?

Other concerns:

1. Y-axis parameter not showed in Fig. 4c.
2. The manuscript would be improved by someone going through and correcting the grammar.

REVIEWER COMMENTS

**Reviewer #A (Remarks to the Author): Expert in computational cancer genomics and**
**structural variants**

----- Overall evaluation -----

*The presented manuscript focuses on the study of 170 whole-genome gastric cancers (GC).*
*The authors analyze GC structural variants (SVs) and found patterns that lead to the*
*identification of 6 rearrangement signatures (RSs). By investigating the impact of VSs, the*
*authors identified 27 SV-hotspots as novel candidate GC drivers.*

*The study further analyzes the profiles of SVs taking into account SV length and allele*
*frequencies, to provide insight into their contribution to GC affected oncogenes and their*
*involvement in extra-chromosomal DNA linked to specific RSs. The manuscript provides*
*valuable new data of whole-genome GC. It creates interesting hypotheses about the active*
*SRs in GC and the mechanism of SV oncogene activation in GC. However, this reviewer*
*has some comments that the authors must address to improve the quality and the rigor of*
*the manuscript.*

**We appreciate the positive comments.**

----- Major comments -----

*1. In the section 'Recurrent SV hotspots represent new GC driver candidates', attempting*
*to detect identify GC-driver SVs the authors state that they focused the analysis on two SV*
*types (Inversion and tandem-duplication) and based on 4 ad hoc conditions they identify 27*
*SV hotspots region as driver candidates. However, in Supplementary Fig. 5 the authors*
*show examples of results from translocation and deletion in know oncogenes. This appears*
*a major incongruence in the result based on the authors' claim that the analysis was*
*focused on Inversion and tandem-duplication.*

*Moreover, this reviewer noticed that the SV calling found 22179 (45%) deletions and 7112*
*(14%) translocations. The prediction of the genomic region of SV hotspots candidate driver*
*should be systematically presented by accounting for each SV type. The identification of a*
*region of SV hotspots as driver candidates with the exclusion of highly prevalent SVs type,*
*or the emphasis on specific SVs type warrants further in-depth explanations. They also can*
*perform statistical analyses to provide the significance of these regions as the driver.*

**Thank you for the critical comment, and sorry for the confusion. We searched for the hot**
**spots for each of the four SV types (Supplementary Figure 3). However, we did not find**
**any candidates for novel gastric drivers in the deletion and translocation types (Additional**
**Data Table 1). To avoid missing significant SV concentrations on GC drivers, we analyzed**
**genomic hotspots on SV groups separated by their effect independent of SV type: one**
**disrupts gene structure, and the other amplifies oncogene, as discussed in the text (page 6,**
**lines 20-23) and Supplementary Figure 4. We were unable to find a new driver for the**
**former, so we have described the results by focusing only on the latter. To clarify this**
**uncertainty, we have added a list of the former type of hotspots as Supplementary Table 5**
**and corrected the text (Page 7, lines 2-6).**

*It will be informative if the manuscript gives complete information about the DEL and TRA*
*for each hotspot region in table 5.*

**We added the counts of DEL and TRA in Supplementary Table 6.**

*2. Gene fusion is of the utmost importance in the cancer genome. Conversely, the*
*manuscript scatters the mentions of genes fusion seemingly as supportive examples. The*
*study implements a new SV caller and there is the availability of RNA-seq therefore the*
*study should provide in the result a particularly comprehensive description of their gene*
*fusion finding no limited but including the SVAF, the breakpoint location (specific exon,*
*intron, or promoter).*

Thank you for the exciting comment. We agree with the reviewer that combining our SV
detection pipeline and RNAseq data can efficiently detect fusion genes in gastric cancer
genomes. We have verified candidate fusion genes in our SV caller by analyzing RNAseq
data as suggested by the reviewer. We validated 54 fusion transcripts in 62 cases where
both WGS and RNAseq data were available. We have included detailed information
(SVAF, the breakpoint location, and annotation) of these fusion events in **Supplementary**
**Table 4** and added this description above in the text (**Page 6, lines 17-20**).

*3. They present a new pattern for SV signature showing its value on the analysis but,*
*having a strong reference of SV signatures resulting from the recent publication of PCAWG*
*project (Li, Yilong et al. “Patterns of somatic structural variation in human cancer*
*genomes.” Nature, 2020), they should establish an in-depth comparison pointing to the*
*strength and weakness of their SV signature proposal.*

We appreciate this critical comment. We compared the methodology and the detection
property of SV signature between the PCAWG project and our method and summarized it
in **Supplementary Table 7**.

The PCAWG method uses 69 classifications based on exhaustive statistics of SV data from
38 cancer types (2,658 cases) to describe the SV features and extract signatures using NMF.
This method assumes that clustered SVs arise from a single DNA replication error and its
subsequent processes, and clustered SVs are classified by their rearrangement pattern of a
chromosome. The resulting signatures are a good reflection of the SV generation process.
Although this is an excellent method, the only weakness is that a portion of clustered SVs
that have irregular rearrangement patterns are not classified into any clustered SV category,
but into distribution-independent classes. Also, it is difficult to apply this method to in-
house data is difficult because there are no publicly available tools to determine
rearrangement patterns of clustered SVs.

We made no assumption about biological processes in SV classification and used the
categorized scheme based on the one proposed by Nik-Zainal (Nik-Zainal, S et al. Nature
2016).

This scheme consists of a minimal set of SV features: SV type, size, and distribution. Thus,
the signatures extracted from this scheme have a weakness in that they are indirectly related
to biological mechanisms. On the other hand, this simple framework has been widely
accepted, which has become an advantage, making it easy to compare with previous
research for different types of cancer. Our classification method, modified with additional
categories of small SVs, transposition, and chromatin state, extracted six rearrangement
signatures, including newly discovered combinations of SV features. We have included a
summary of the description above in the text (**page 8, lines 13-18**).

*4. In the section ‘Extrachromosomal DNA-driving SVC generates driver oncogene
amplicons in GC’, the authors claim that SVCs with amplicon types are enriched in GC
oncogenes, suggesting that they were positively selected for GC development. That
statement must be supported by the corresponding statistical analyses, which should also
include the confirmation that the enrichment is not due to SV signature thus, indeed it is a
sign of their potential as positively selected regions.*

We appreciate this significant comment. Fisher's test was performed on the frequency of
oncogenes between genomic regions with amplicon-associated SVCs and the rest of the
genome and confirmed a significant enrichment of oncogenes at these SVCs (P=0.01778).
Our method defines SV clusters independently of the SV signature extraction process, and
the enrichment of oncogenes in the amplicon-associated SVC is not due to the specific
signature process. However, as the reviewer suggested, this does not mean that this process
is “positively selected.” Detecting the selection signal in the SV process would be more
complicated. We overstated the text, and this description is misleading. Therefore, we have
corrected the text (**Page 15, lines 2-3**)

5. In section ‘Rearrangement signatures and subtypes in GC’, The authors state that “RS4, consisting of large (0.1 to 1.0 Mb) deletions occurred in the active chromatin area” but Figure 2 does not support that statement. There is the same level of RS4 deletion in the inactive chromatin area.

Sorry for the confusion. We have shown the absolute number of events in Figure 2. Since the inactive chromatin genomic region (2,374.4 Mb) is 4.88 times larger than that of the active chromatin region (486.8 Mb), the frequency of events in the inactive regions is higher than that in the active region. To accurately show the SV composition in the two chromatin regions, the SV composition of each RS shown in **Figure 2(a)** was converted to the frequency per 100Mb in each chromatin state and added as **Supplementary Figure 9**. Moreover, information on the source of the epigenomic chromatin annotation was added to the **Methods** section (**Page 18, line 19-23**).

6. The study implements a new method for detecting SVs from WGS without providing any benchmark with the established methods. The manuscript needs a proper benchmarking of the new method providing curves to interpret sensitivity, specificity, precision, and accuracy.

We appreciate this critical comment. We agree with the reviewer’s suggestion and performed a benchmark test using GenomonSV (Chiba K. et al., Bioinformatics 2015, <https://github.com/Genomon-Project/GenomonSV>), which established SVcaller in studies including myeloma (Yoshizato T. et al., Blood 2017), leukemia (Madan, V et al., Leukemia 2016), lymphoma (Yasuda T. et al, Cancer Sci 2020), and pneumocytoma (Yeh, YC. et al, Mod Pathol 2020). The result is shown in **Supplementary Figure 18**, and the detailed test procedure is described in the **Methods** section (**Page 21, line 2-13**).

7. The section ‘PCR validation’ shows incongruent numbers. Initially, the authors state the
validation of 120 predicted somatic SVs. But they precede by saying that they randomly
selected 87 SVs. Therefore, it seems like the first statement is incorrect. Moreover, they
continue by reporting the confirmation of 72 SV breakpoints out of the selected 87 but, it
appears that they report a prediction accuracy of 96% (72/75), which is incorrect. What we
can say from that data is ‘in 72.5% (87/120) of the predicted somatic SVs, the sensitivity or
true positive rate (TPR=TP/P) is 82.8 (72/87)’. This reviewer suggests revisiting the
concepts in this section to provide the correct analysis by checking the validation of
positive and negative predicted SVs.

We apologize for the incorrect description. We have confirmed and corrected the data. The
relevant section of the method has been revised (**Page 5, line 23 and Page 22, line 15 –**
**Page 23, line 2**).

----- Minor comments -----

- 1. Figure 1b does not show clearly the 27 SV hotspot regions identified as driver
candidates.
- 2. Line 457, there is a repeated word: ‘fusion’.
- 3. Supplementary Fig. 9, typo (4 instead of 1) in RSI.
- 4. The manuscript states, “The subtype RS2 contained MSI-positive and EBV-positive cases
(Supplementary Fig. 10a)” but, based on the legend of Supplementary Fig. 10, this graph
does not show MSI-positive cases.

Thank you for pointing out these points. We have corrected these four points as follows.

- 1. We assigned an identifier to each of the 27 SV hotspots shown in the first column of
Supplementary Table 6, and subscripted the identifier to the hotspot indicator (red line) in
Figure 1b.
- 2. “fusionfusion” is the name of a pipeline. We have corrected this (**Page 23. Lines 10**)

- 3. We have corrected the typo in **Supplementary Figure 12** (previously **Supplementary**
- **Fig. 9**).
- 4. We apologize for the missing data. The graph for MSI-positive cases has been added to
- **Supplementary Figure 13(a)** to clarify the text on subtype-RS2.

**Reviewer #B (Remarks to the Author): Expert in gastrointestinal cancer genomics**

*1. The authors used a new computational method to identify SVs in the paper that relies on*
*the VAF of SV (SVAF). It is unclear how the VAF of amplified regions can be estimated*
*robustly especially in cases of amplicons and ecDNA. An example in the supplementary*
*data would illustrate the robustness of using SV VAF in classifying regions with SVs and*
*including highly amplified regions. A correlation plot similar to Fig1a bottom with SVAF*
*and CN as x and y axes would illustrate the relationship as well.*

**We appreciate the very critical comment. As the reviewer pointed out, we agree that it is**
**difficult to determine the VAF of the highly amplified regions and ecDNA. For example, it**
**is difficult to determine the clonality of SVs that occur within ecDNA or amplicon**
**segments. As suggested by the reviewer, we generated a correlation plot of copy number**
**and SVAF (Supplementary Figure 19). We found that there was no specific bias of SVAF**
**in the highly amplified genomic regions, suggesting that the SVAF in the study could**
**adjust for the effect of copy number changes. High SVAF-SV hot spots were detected not**
**only from the amplified oncogene locus, but also from the tumor suppressor gene locus**
**(shown in Supplementary Figure 6 and Supplementary Table 5). These results also**
**support that the SVAF evaluation is effective regardless of the segmental copy number**
**where the SV breakpoints are located. We have added these data and mentioned this point**
**in the text (Page 7, line 2-6).**

*2. The section of SV signatures is very informative as the correlations with other clinical*
*and molecular features provides insight on the molecular subtypes in GC. The addition of*
*chromatin features provided an additional dimension to understand the processes*
*generating the SVs. There are several minor comments on the methodology and*
*presentation of the data. For the SV signature extraction, how many iterations were carried*
*out and were the signatures presented in the manuscript robust? Similarly, the clustering of*
*patients into subtypes can be assessed using consensus clustering to see how stable the*

*clusters are.*

Thanks for the positive comments on our SV signature methods. We ran 50 NMFs with
iteration=100 and found that all of them yielded RS sets almost identical to those in this
paper (the sum of the pairwise cosine distances corresponding RSs is less than 0.0001).
Therefore, we believe that our signature extraction result is reproducible and robust.

As suggested by the reviewer, we re-clustered the GC 170 cases using a consensus
clustering algorithm with Ward2 linkage (Wilkerson, MD & Hayes, DN, Bioinformatics
2010). The clustering output (RS subtype classification) was almost identical to the
previous one, with the sole exception of pfg038, which changed from subtype RS1 to
subtype RS2. Because these clusters are more robust than the original ones, we replaced our
original one and presented this new result in **Figures 2a and 2b**.

3. *The observation of a correlation between RS1 and the BRCA phenotype is clinically*
*important. The addition of SNV and SV signatures provides a reliable way to identify*
*GC patients with the BRCA phenotype with confidence especially with the addition of*
*the inactive chromatin features.*

Thanks for the positive comment. We agree that adding SNV signature data would be more
reliable for identifying the cases with BRCA phenotype. We have verified that RS1, and
SBS3 and ID6, which are both associated with the BRCA phenotype, are significantly
correlated (**Supplementary Figure 10**).

*What does the SV signatures look like without the addition of the inactive chromatin*
*features and how would the patients with the BRCA phenotype be grouped otherwise in*
*other subgroups? Also, could the authors comment on how many cases would be missed if*
*only the SV analysis is carried out, without the addition of the chromatin features?*

Thank you for this important question regarding the clinical significance of our SV
signature extraction. To answer this question, we performed an NMF analysis without the
chromatin state data. This generated seven rearrangement signatures (RS') (**Additional**
**Data Figure 1a**). These included three RSs (RS3', RS4', and RS5') similar to RS3, RS4,
and RS5. The remaining four (RS1', RS2a', RS2a', and RS6') were reconstructed from RS1,
RS2, and RS6. We then examined whether eight cases with the BRCA phenotype (defined
by SBS3 and ID6 contributions ≥ 0.3 and indicated by the asterisk in **Supplementary**
**Figure 10** and **Additional Data Figure 1b**) were enriched in the subtype RS1'. Five were
in the subtype RS1', but the remainder (3 cases) were in the subtype RS2a'. Although we
need more cases to verify the classification power of the BRCA phenotype in the gastric
cancer cohort, our analysis suggests that SV analysis without chromatin features may be
less effective in identifying the BRCA-enriched subtype. As suggested by the reviewer,
combining the SNV data would strengthen the stratification of these cases, as reported in
the breast cancer cases (Davies H. et al., Nat Med. 2017).

*4. It is important to match extracted SV signatures to reference signatures for the better*
*comparison of signatures and prevent mis-identification of signatures. What are the*
*signatures identified in this study and how do they relate to SV signatures in*
*<https://signal.mutationalsignatures.com/> ? Are there new SV signatures that are previously*
*not discovered?*

Thank you for the very critical comment. We agree that comparison of our SV signatures
(RSs) with the previously reported reference rearrangement signatures is quite beneficial to
evaluate the validity of our method.

Signal has extracted the SV signatures in the “stomach cancer” genomes (as described in
Signal) and reported them as RefSig R (RSR, hereafter) 1, 2, 3, 4, 6a, 6b, 7, and 9. We
deconstructed our SV dataset of GC 170 cases with this set of stomach cancer RSRs
(**Supplementary Figure 10d**) and compared their contributions with the best-

corresponding RS (**Supplementary Figure 10e**). The comparison revealed significant
correlations between RS3 and RSR1 ($\tau=0.60$, $P<2.2E-16$), RS4 and RSR7 ($\tau=0.64$,
$P<2.2E-16$), and RS6 and RSR6b (0.55, $P<2.2E-16$). Also, RS1 and RSR9 ($\tau=0.20$,
$P<2.8E-04$) and RS5 and RSR4 ($\tau=0.41$, $P<2.0E-12$) show weak correlations, whereas
RS2 has no correlated RSR in the stomach cancer set. This result did not change when the
dataset was deconstructed with another set of RSRs consisting of the RSRs of the stomach
set and RSR10 (the most similar to RS2), RSR16 (to RS6), and RSR16 (to RS5)
(**Additional Data Table. 2** and **Additional Data Figure 2**).

RS2 contains the co-occurrence of small-sized deletions and tandem duplications, and this
signature may be a newly discovered one. We have added this to the text (**Page 8, lines 23**
**– Page 9, line 5**).

*5. The NAHRD signature closely resembles another tandem duplication signature known to*
*be associated with CCNE1 amplifications (RS1 in signal). An analysis of tandem*
*duplication sizes would be of interest to identify cases with small tandem duplications that*
*might be missed in the signature analysis.*

Thank you for the critical comment. As the reviewer pointed out, our NAHRD (RS3)
signature is similar to the RSR1 in Signal (as described in the response above, please see
**Supplementary Figure 10** and **Additional Data Table. 2**). We also found that our RS3 is
associated with *CCNE1* amplification. The Signal pipeline also detected a unique SV
signature (RSR3) enriched for small-sized tandem duplications (smallTDs). However, this
signature was not extracted in our analysis. On the other hand, the results of deconstructing
the SV data of GC 170 cases SV data with the stomach cancer set of RSRs showed that out
of forty-two cases with 10% or more contribution of RSR3, 69% (29/42) of them were in
the RS2 subgroup and 14% (6/42) in the RS2/RS6 subgroup (**Supplementary Figure 10d**).
As described above, RS2 contains a co-occurrence of small deletions and tandem
duplications. Therefore, by including small-sized SVs in the categorization, the smallTDs
signature was mainly integrated into the RS2, a new signature in our analysis.

6. The grouping of patients according to SV signatures shows how diverse GC is as a disease and it would be useful to groups or subtypes better and with more informative names. St.RS2, StRS6 and StRS2/6 seems very distinct classes given the differences in SNV mutation counts and the presence of a MSI cluster within RS2. Can the MSI patients be split out as a cluster?

Thank you for this positive comment. We fully agree with the reviewer that SV signature analysis would identify novel or unique molecular subtypes in gastric cancer. We have established a more robust molecular classification in response to this reviewer's #2 comment. In this new classification, we recognized a group of samples within stRS2, in which the contribution of RS2/RS4 is greater than 85% (**Figure 2(b)**). This cluster shows a low frequency of SV ($p=1.219e-15$) and is enriched in MSI (14.3%, 4/28) and hypermutation cases ($p=1.54*10E-2$, $p=6.93*10E-4$, respectively). We have included this result in the text (**Page 10, line 16-18**).

Does stRS6 and stRS2/6 have similar features and are the clusters separate clusters meaningful for interpretation? Clustering based on Ward.D2 might give a different clustering outcome.

Thanks for the comment. We tried the clustering samples with all the linkage types provided by Consensus Clustering Plus (average, complete, mcquitty, median, wardD, and wardD2). However, none of the clusterings merged stRS6 and stRS2/6 into one cluster. As these two subtypes have significant differences in the histological classification and the number of SVs, we consider them to be distinct molecular subtypes.

7. The classification of SVs into classes according to the biological mechanism is challenging. This is especially in complex rearrangements where a large number of SVs are within a genomic region, where it is more challenging to identify each biological

*mechanism. It is unclear how the self-joining amplicons and assembled amplicons are*
*different from ecDNA and if the additional categories provide additional information on*
*the differences of the processes generating these SVs. Are there differences in the*
*proportions of inversions, breakage-fusion bridges or other features in these complex SV*
*classes that might point to different mechanisms?*

Thank you for this inspiring comment. We agree with the reviewer's comment. As the
reviewer pointed out, our self-joining and assembled amplicons contain ecDNA; however,
the difference between the two is unclear. As suggested by the reviewer, we compared the
proportions of fold-back inversions, which are associated with breakage-fusion bridges
(BFBs), and nested tandem duplications (Nested-TDs) among them (**Supplementary**
**Table 10**). As expected, fold-back inversions were significantly observed in the BFB-type
amplicons. Nested-TDs were significantly higher in the NAHRD type and in the self-
joining and assembled amplicon classes. Therefore, we could not find a specific molecular
property that links unique molecular mechanisms in ecDNA. As previously reported
(Bergstrom, E.N. et al. Nature, 2022), ecDNA might be more susceptible to APOBEC-
associated mutagenesis. Thus, the enrichment of specific mutation patterns (*kataegis* and
*omikli*, in response to Reviewer#3 comment 2) or epigenetic modifications (Wu S. et al.
Nature 2019) might be the key. This would be a goal of our future study. We have added
this description above in the text (**Page 12, lines 19-21**).

8. Figure 3G needs to be refined as the lines are obscuring the copy number profiles.
Authors can change the transparency value of the lines to show the profile better.

Thank you for the comment. As suggested by the reviewer, we have revised **Figure 3G** by
re-selecting a representative case (#pfg166 instead of #GC004) where the change in CN
ratio at the SV breakpoint can be more clearly observed.

9. ecDNA and HSRs are difficult to differentiate without FISH staining done

*experimentally. Have the authors attempted to identify other cases with reintegration and if*
*the SVAF provides a way to identify such cases?*

We appreciate the very insightful and interesting comment. As the reviewer pointed out, it
is difficult to distinguish ecDNA and HSRs without FISH. As the reviewer suggested,
However, as the reviewer suggested, it is possible to infer an SV trace generated by ecDNA
reintegration (\approx HSR) from information such as SVAFs, the location of SV breakpoints, and
copy number variation there. We therefore set the following three criteria:

(1) One of the two breakpoints of an SV is detected in the region of a self-joining amplicon
or an assembled amplicon, and the other is detected from a non-SV clustered region,

(2) The breakpoint of a non-SV clustered region was not accompanied by a distinct copy
number change (dCN required less than 0.5) before and after its coordinate,

(3) The size of an SV was more than 2 Mb, or the type was translocation-SV.

157 SVCs and 471 SVs were extracted as candidates for reintegration that met these
conditions (**Supplementary Table 11**). To validate this assumption, we additionally
performed a FISH analysis on three cases where we predicted the presence of ecDNA and
HSRs. We successfully identified scattered bright spots indicative of ecDNA, and the HSR
in all cases (**Figure 4b** and **Supplementary Figure 16b-d**). We have added this description
above in the text (**Page 13, lines 19-21**).

**Reviewer #3 (Remarks to the Author): Expert in gastric cancer genomics**

Major Concerns:

*1. Authors totally analyzed 170 gastric cancer (GC) genome sequencing WGS data,*
*identified out 49,059 SVs. According to the recent pan-cancer analysis of whole genomes*
*research, I was wandering about that weather the authors took diffuse hypermutation*
*(omikli) and longer events (kataegis) signatures into concern.*

We appreciate this inspiring comment. SigProfilerClusters (Bergstrom EN, et.al.,
Bioinformatics, 2022) identified 1,356 *kataegis* and 31,541 *omikli* regions in our 170 WGS
data. We have presented the data in **Supplementary Table 4** in additional columns named
'Kataegis' and 'Omikli', and they are also shown in **Figure 2(b)**.

*2. On the other hand, is there any omikli or kataegis signature in the ecDNA author*
*identified?*

We appreciate this critical comment. Based on the reviewer's suggestion, we analyzed the
correlation between the number of *kataegis* events and RS contribution in our cohort. We
found a significant positive correlation between *kataegis* and RS6 (Kendall $r=0.345$,
$p=3.68e-08$). We also found a negative correlation between the number of *omikli* and RS4
($r=-0.261$, $p=2.33e-06$) (**Supplementary Table 8**). Further examination of the RS subtypes
revealed a high frequency of *kataegis* in stRS3 and stRS6 ($p=3.00E-10$ and 0.031 ,
respectively) and a high frequency of *omikli* in stRS1 and stRS3 ($p=0.021$ and 0.046 ,
respectively) (**Table 1**). We have included these results in the text (**Page 9, line 18-23** and
**Page 10, line 7**).

We then examined the frequency of *omikli* and *kataegis* in all six types of SV clusters
defined in our study. We found that *kataegis* and *omikli* were enriched in the BFBC, self-
joining, and assembled amplicon regions compared to CFS, L1, and NAHRD
(**Supplementary Figure 17**). The two highest frequencies were observed in the self-joining

and assembled amplicons, which are associated with RS6. Therefore, as reported in a
previous study (Bergstrom, E.N.et al. Nature, 2022), *omikli* and *kataegis* signatures were
enriched in ecDNA. We have included these results in the text (**Page 13, line 21-24**).

*3. What's ration of the ultra-large-scale SVs that connect intra-cluster SVs and distantly*
*located genomic regions corresponded to the reintegration of ecDNA into the genome in*
*the cohort? not showed in figures.*

Thank you for pointing out an important point. Of the total 49,059 SVs detected, 863 SVs
connected two or more SV cluster regions. The ultra-large SVs possibly that may be
involved in chromosomal reintegration were identified by the following two additional
conditions:

(1) One of the two breakpoints of an SV is detected in the region of a self-joining amplicon
or an assembled amplicon, and the other is detected from a non-SV clustered region,
(2) The breakpoint of a non-SV clustered region was not accompanied by a distinct copy
number change (dCN required less than 0.5) before and after its coordinate. Based on these
criteria, 471 SVs were identified (**Supplementary Table 11**). The ratio between these two
types of SVs is approximately 1.8 (863:471). In addition, we have added the results of
FISH validation for three candidates for reintegration as **Supplementary Figure 16b-d**,
and descriptions of them in the text (**page 13, line 17-21**).

*4. Somatic mutations filters were based on RNA-seq, for low-frequency variants may not be*
*adequately observed in genomic DNA but enjoy high read support in RNA if the*
*corresponding genes are highly expressed. So, is there any additional somatic variant-*
*calling pipelines can be contributed to the analyses?*

We appreciate this comment. As the reviewer pointed out, transcriptional data are helpful
for detecting low-VAF mutations where enough WGS coverage is insufficient. However, it
only covers the transcribed genomic region, and the quality of the mutation call may differ

between cases with and without transcription data. Furthermore, when these mixed data are
used for downstream mutational signature analysis, the different sensitivity of mutation
detection would confound the interpretation of the results. Unfortunately, only 62 out of
170 cases in this study have corresponding RNAseq data in this study. Therefore, we
detected somatic mutations solely by analyzing whole genome sequencing.

---

Other concerns:

1. *Y-axis parameter not showed in Fig. 4c.*

2. *The manuscript would be improved by someone going through and correcting the*
*grammar.*

Thank you for these comments. We have corrected these points and the revised manuscript
has been proofread by a native speaker.

In addition to the above revisions requested by reviewers, the following changes were made
in our script:

* Adopting the Consensus Clustering results, one case (pfg058) was classified into stRS2
from stRS1. For this change, we recalculated statistics for the properties of the subtypes
RS1 and RS2.

* In the SVC profile classification scripts, the condition filters were not working correctly
in some cases. Therefore, we fixed the script error, and 3,457 SVCs were redefined. And
we updated Fig.3a, Fig.3e, Fig. 4c-h, Supplementary Fig.14-15, and 16th column of
Supplementary Table 4, which were processed from the SVC definition.

* The numerical values in Fig. 4 (f,g) were changed from the ratio to the number of
observed regions. This change prevents confusion in discussing the proportions of each
SVC in amplified clusters on page 14.

* In Fig4(h), we corrected the confusion of color codes indicating NAHRD (green) and
BFBC (light blue).

**Additional Data legend**

**Additional Data Figure 1 Rearrangement signatures extracted without chromatin states**

a. Seven rearrangement signatures were extracted using the 40 features that abolish active or
inactive distinctions of chromatin status from the 80 features used in our study.

b. Unsupervised hierarchical clustering of 170 cases based on a set of the 40 features RSs
contribution (top) and the contribution of SBS, ID, and DBS signatures calculated using the
deconstructSigs2.

The asterisks on the RSs contribution plot indicate cases with BRCA phenotypes defined as
their SBS3 and ID6 contribution accounting for 30% or more.

**Additional Data Figure 2 GC case deconstruction by a modified set of reference
signatures for rearrangement (RSR) proposed by Signal.**

The SV dataset of GC 170 cases was deconstructed by a set of 11 RSRs, which consisting of
8 RSRs (1, 2, 3, 4, 6a, 6b, 7, and 9) already proposed by Signal for “stomach cancer”, and 3
RSRs (10, 16, and 18) similar to the RSs extracted in our study. RSR10 is the reference
signature with the closest cosine similarity to RS2, RSR16 to RS6, and RSR18 to RS5 (see
Additional Data Table 2).

a. The top row is the hierarchical clustering of 170 cases by RS contribution, taken from
Figure 2b. The lower row shows the aligned RSR contribution in each case aligned.

b. Correlation plots between an RS and its corresponding RSR. A red colored plot indicate
that the RS-associated RSR was not included in the Signal “stomach cancer” RSR set.

**Additional Data Table 1. Genomic hotspots of Deletion and Translocation.**

Hotspots were defined as genomic regions where high SVAF-SV (SVAF>0.4) was detected
in five or more cases per 0.1 Mb, excluding deletion clusters in common fragile sites and
translocation clusters due to active L1-transposon.

**Additional Data Table 2. Pairwise comparison of cosine similarity between six
rearrangement signatures (RSs) and twenty-one reference signatures for
rearrangement (RSRs) proposed Signal.**

The six RSs extracted in our GC studies included 80 features, including active or inactive
discrimination of chromatin states. For comparison with the RSRs composed of 32 features
consisting of four SV types, five SV sizes, and two distribution states, we reconstructed the
RSs using the same scheme of the 32 features similar to the Signal.

REVIEWERS' COMMENTS

Reviewer #1 (Remarks to the Author):

The authors have responded satisfactorily to all comments and have updated the manuscript accordingly. The current version of the manuscript has better explained the result by adding corresponding statistical tests and more precise conclusive statements. The manuscript now has additional and updated tables and figures that improve the clarity of the results. The updated text expands on the results on gene fusion detection. In the current version, the authors have compared their method of detecting SV signatures with the methodology used in the PCAWG project. The current version also properly compares its SV detection methodology with an alternative method (GenomonSV). Minor comments have been corrected too.

Reviewer #2 (Remarks to the Author):

The authors have provided substantial evidence on how the different molecular processes interact in gastric cancer by combining mutational signatures of different modalities with rearrangement processes underlying large scale rearrangements. Of note, the additional of chromatin status as a feature to separate molecular processes has allowed the identification of BRCA related homologous recombination as dominant process in gastric cancer.

In addition, the development of an SV caller (CallallSV) has provided additional tools to detect a variety of rearrangements and allowed for both simple and complex events to be identified. A strength of this study is the annotation of the rearrangement events identified and additional support by both RNA-seq data and experimental validation. The authors have provided an additional approach to identify integration of ecDNA events into the genome and supported the findings well with FISH validation of the events.

REVIEWER COMMENTS

**Reviewer #1 (Remarks to the Author):**

*The authors have responded satisfactorily to all comments and have updated the*
*manuscript accordingly. The current version of the manuscript has better explained the*
*result by adding corresponding statistical tests and more precise conclusive statements.*
*The manuscript now has additional and updated tables and figures that improve the clarity*
*of the results. The updated text expands on the results on gene fusion detection. In the*
*current version, the authors have compared their method of detecting SV signatures with*
*the methodology used in the PCAWG project. The current version also properly compares*
*its SV detection methodology with an alternative method (GenomonSV). Minor comments*
*have been corrected too.*

**We appreciate your kind review of our study on gastric cancer. Your comments are to the**
**point, and the process of considering your suggestion improved the quality of our research**
**articles.**

**Reviewer #2 (Remarks to the Author):**

*The authors have provided substantial evidence on how the different molecular processes*
*interact in gastric cancer by combining mutational signatures of different modalities with*
*rearrangement processes underlying large scale rearrangements. Of note, the additional of*
*chromatin status as a feature to separate molecular processes has allowed the*
*identification of BRCA related homologous recombination as dominant process in gastric*
*cancer.*

*In addition, the development of an SV caller (CallallSV) has provided additional tools to*
*detect a variety of rearrangements and allowed for both simple and complex events to be*
*identified. A strength of this study is the annotation of the rearrangement events identified*
*and additional support by both RNA-seq data and experimental validation. The authors*
*have provided an additional approach to identify integration of ecDNA events into the*
*genome and supported the findings well with FISH validation of the events.*

**We appreciate your kind review of our study on gastric cancer. Your comments are to the**
**point, and the process of considering your suggestion improved the quality of our research**
**articles.**